# Improving Monte Carlo Tree Search for Symbolic Regression

**Zhengyao Huang**[*]
2301213083@stu.pku.edu.cn

**Daniel Zhengyu Huang**[†]
huangdz@bicmr.pku.edu.cn

**Tiannan Xiao**[‡]
alxeuxiao@pku.edu.cn

**Dina Ma**[‡]
madina@huawei.com

**Zhenyu Ming**[‡]
mingzhenyu1@huawei.com

**Hao Shi**[§]
shih22@mails.tsinghua.edu.cn

**Yuanhui Wen**[‡]
wenyuanhui@huawei.com

## Abstract

Symbolic regression aims to discover concise, interpretable mathematical expressions that satisfy desired objectives, such as fitting data, posing a highly combinatorial optimization problem. While genetic programming has been the dominant approach, recent efforts have explored reinforcement learning methods for improving search efficiency. Monte Carlo Tree Search (MCTS), with its ability to balance exploration and exploitation through guided search, has emerged as a promising technique for symbolic expression discovery. However, its traditional bandit strategies and sequential symbol construction often limit performance. In this work, we propose an improved MCTS framework for symbolic regression that addresses these limitations through two key innovations: (1) an extreme bandit allocation strategy tailored for identifying globally optimal expressions, with finite-time performance guarantees under polynomial reward decay assumptions; and (2) evolution-inspired state-jumping actions such as mutation and crossover, which enable non-local transitions to promising regions of the search space. These state-jumping actions also reshape the reward landscape during the search process, improving both robustness and efficiency. We conduct a thorough numerical study to the impact of these improvements and benchmark our approach against existing symbolic regression methods on a variety of datasets, including both ground-truth and black-box datasets. Our approach achieves competitive performance with state-of-the-art libraries in terms of recovery rate, attains favorable positions on the Pareto frontier of accuracy versus model complexity. Code is available at https://github.com/PKU-CMEGroup/MCTS-4-SR.

## 1 Introduction

Symbolic regression (SR) is a machine learning methodology that aims to discover interpretable and concise analytical expressions to represent **data-driven relationships** or **desired functional objectives**, without assuming any predefined mathematical form. Unlike traditional regression techniques, which optimize parameters within fixed model structures, symbolic regression simultaneously searches for both the functional form and its parameters by exploring a combinatorial space. The resulting expressions are often simple and human-readable, which has supported their utility in

---

[*]Center for Machine Learning Research, Peking University, Beijing, China.

[†]Corresponding author; Beijing International Center for Mathematical Research, Center for Machine Learning Research, Peking University, Beijing, China.

[‡]Huawei Technologies Ltd., Beijing, China.

[§]Department of Mathematical Sciences, Tsinghua University, Beijing, China.

39th Conference on Neural Information Processing Systems (NeurIPS 2025).

scientific discovery [1, 2]. In many cases, SR-derived models align with domain-specific laws, enabling their application across domains such as finance [3], materials science [4], climatology [5], and healthcare [6]. Additionally, the combination of computational efficiency and model simplicity makes SR well-suited for time-sensitive scenarios like real-time control systems [7], where rapid execution and operational transparency are essential.

Symbolic regression is a combinatorial optimization problem, as expressions can be represented as binary trees or symbolic sequences (see Section 2.1). Finding an optimal expression entails searching both the tree structure and its parameters—proven to be an NP-hard task [8]. To address this, numerous heuristic and approximate methods have been proposed and benchmarked [9, 10]. Among these, **genetic programming** (GP) remains the most widely adopted approach: GP evolves a population of candidate expressions via mutation, crossover, and selection to optimize data fit [11–15]. However, GP frequently produces overly complex formulas and exhibits high sensitivity to hyperparameter settings [16]. More recently, learning-based methods have been proposed to exploit structural regularities—such as symmetry, separability, and compositionality—in symbolic expressions[17]. **Deep neural networks** can be trained to generate candidate formulas directly [18–20], but they often excel only at data-fitting tasks and struggle to generalize beyond the training distribution. An alternative is to cast SR as a sequential decision-making task and apply **reinforcement learning** (RL). Inspired by breakthroughs in games like Go [21, 22], RL-based SR leverages policy-gradient or recurrent-network agents to construct expressions token by token [16, 23, 24]. Likewise, **Monte Carlo Tree Search** (MCTS) has been adapted to navigate the space of symbolic programs more effectively [25–29].

While symbolic regression can be reformulated as an RL problem involving the determination of an optimal sequence construction policy, it differs from conventional RL in two critical ways: (1) the objective is to identify the globally optimal state (i.e., the highest reward) rather than to maximize the expected cumulative rewards, and (2) the action space is not fixed but can be designed to enable flexible state-jumping mechanisms, bypassing traditional stepwise transitions. The goal of this work is to analyze and enhance the application of MCTS within the framework of symbolic regression. Our key **contributions** include:

- Formulating an extreme-bandit allocation strategy, analogous to the Upper Confidence Bound (UCB) algorithm, and derive optimality guarantees under certain reward assumptions.

- Introducing evolution-inspired state-jumping actions (e.g., mutation and crossover, as used in genetic programming) to shift the reward distribution in MCTS towards higher values, improving both exploration and exploitation.

- Developing a refined MCTS framework for SR that (1) attains competitive performance against state-of-the-art libraries and (2) yields new insights into applying RL methods to combinatorial optimization.

### 1.1 Related Work

**Monte Carlo Tree Search for Symbolic Regression.** Inspired by the success of AlphaGo [21, 22], MCTS has been explored in symbolic regression. An early effort [26] employs a pretrained actor-critic model for the value function within the UCB scheme. Later, [27] improves MCTS with module transplantation, which manually adds symbolic sub-expressions (e.g., $x^4$) from high-reward states in the leaf action space. Another work [28] refines the UCB scheme by incorporating transformer-guided action probability assessment following AlphaGo-inspired UCB scheme [21, 22]. More recent work [29] introduces a hybrid approach that alternates between genetic algorithms and MCTS, where MCTS results are used to initialize GP, and GP's results are leveraged to train a double Q-learning block within MCTS. A common trait across these methods [27–29] is the emphasis on exploiting high-reward actions: [27] and [29] adopt an $\epsilon$-greedy selection strategy, whereas [28] modifies the AlphaGo-inspired UCB formula by replacing the average with the maximum observed reward.

**Bandit-Based Allocation Strategies.** The Upper Confidence Bound (UCB), originally proposed for multi-armed bandits with uniform logarithmic regret [30], balances exploration and exploitation in MCTS [31]. Integrating UCB into MCTS improves planning performance [32]. However, whereas UCB minimizes average regret, tasks like symbolic regression seek the single best outcome. To address this, researchers have studied the extreme-bandit setting [33–37] and applied it to combinatorial challenges such as Weighted Tardiness Scheduling. Here, we propose an extreme-bandit allocation

rule with optimal guarantees under certain reward distribution assumptions, and embed it within MCTS for symbolic regression.

**Hybrid Methods for Symbolic Regression.** Symbolic regression is an NP-hard problem [8], often tackled using heuristic methods like GP and RL, which aim to efficiently explore the expression space under limited computational budgets. GP evolves a population of candidate solutions through biologically inspired operations such as mutation and crossover [11]. This process can be modeled as a finite-state Markov process, where convergence relies on ergodicity [38, 39]. However, the effectiveness of GP depends on a careful balance between exploration and exploitation, which in turn is sensitive to hyperparameter tuning. RL-based approaches, such as MCTS and policy gradients [40], offer an alternative by learning policies to construct expressions symbol by symbol. Notably, [16, 41] apply risk-aware policy gradient methods [42, 43] using recurrent neural networks to guide expression generation. These methods tend to strike a better exploration-exploitation balance, though they may suffer from inefficiency due to their stepwise construction of expressions. To overcome these limitations, recent works [23, 24, 29] have proposed hybrid methods that combine RL and GP in a modular fashion, alternating between them to enhance overall performance. Building on this idea, the present work incorporates biologically inspired state-jumping operations—namely mutation and crossover—into MCTS. This hybrid strategy not only improves exploration efficiency but also aims to enhance the interpretability and effectiveness of the search process.

## 2 Background

### 2.1 Symbolic Regression

Symbolic regression is a combinatorial optimization problem that aims to find an expression $f$ minimizing a nonnegative objective functional:

$$\min_f \mathcal{L}(f) \tag{1}$$

In data-fitting scenarios, $\mathcal{L}(f)$ measures the discrepancy between predictions and observations. A common choice is the normalized root mean square error (NRMSE):

$$\text{NRMSE}\big(f; (x_i, y_i)_{i=1}^n\big) = \sqrt{\frac{\sum_{i=1}^n (f(x_i) - y_i)^2}{\sum_{i=1}^n (y_i - \bar{y})^2}}, \tag{2}$$

where $\bar{y} = \frac{1}{n} \sum_{i=1}^n y_i$ is the mean of the observed values. Beyond data fitting, symbolic regression can also generate expressions with desired analytical properties, such as control functions. The expression $f$ is constructed from a predefined symbol set, including arithmetic operators, elementary functions (e.g., $\sin$, $\cos$), constants, and variables—making the search space inherently combinatorial.

Expressions are commonly represented as binary trees, with internal nodes as operators or functions and leaves as constants or variables. These trees are often traversed in pre-order to yield sequential forms suitable for learning algorithms (see Figure 1).

### 2.2 Markov Decision Process

Due to the sequential nature of expression representation, the task of finding the optimal expression can be formulated as a Markov decision process (MDP). The state space $\mathcal{S}$ contains valid sequences of symbols, which may represent incomplete expressions. The action space $\mathcal{A}$ consists of a set of predefined symbols. Taking an action $a \in \mathcal{A}$ at a state $s \in \mathcal{S}$ deterministically transitions to a new state $s' = \{s, a\}$, where the action $a$ is appended to the sequence.

The reward function $r(s, a, s')$ is defined as zero if the resulting state $s'$ is still an incomplete expression. However, if $s'$ forms a complete and valid expression, the reward is given by $\frac{1}{1+\mathcal{L}(s')}$, where $\mathcal{L}$ denotes the objective functional defined in (1) and (2). This reward lies within the interval $(0, 1]$. Complete expressions are treated as terminal states in this process.

In practice, the length of the expression sequence or the maximum depth of the expression tree is typically bounded in order to limit the size of the state space. The goal is to find a sequence of actions that, starting from the empty state $s_0 = \varnothing$, leads to a complete expression that maximizes the reward. When interpreted as a pre-order traversal, this sequence yields the optimal expression. The detailed modeling procedure can be found in Appendix D.

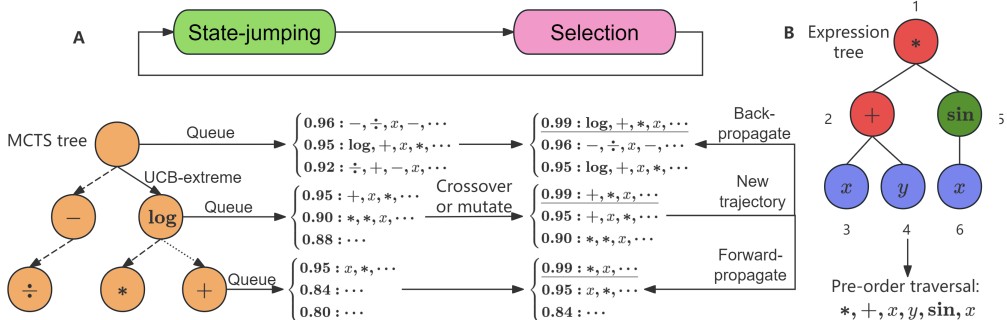

Figure 1: **A**. Schematic of State-jumping Actions. Each MCTS node maintains a queue of top-$N$ trajectories passing through it. Before the standard selection step, a probabilistic State-jumping operation may be applied: randomly mutating or crossing trajectories from the queue to generate new states at the same node. This bypasses conventional RL sampling by directly introducing promising candidates. In this example, the UCB-extreme criterion (7) selects a second-layer node ($\log$), which then generates new child states ($+, *, x, \cdots$) with a higher reward (0.99). Bidirectional propagation ensures that both parent and child queues are updated. **B**. Illustration of converting a binary expression tree into a pre-order sequence. The example expression is $(x + y) * \sin(x)$.

## 2.3 Monte Carlo Tree Search

MCTS [32] is a sample-based planning algorithm that incrementally builds an asymmetric search tree, where each node represents a state (e.g., a valid symbol sequence), and edges correspond to actions. Each node maintains visit counts and estimated action values to guide the search. MCTS proceeds through four phases:

In the *selection* phase, the tree is traversed from the root using the UCB criterion until a leaf with unvisited children is reached. At node $v$ for state $s$, the selected action is

$$a = \operatorname{argmax}_a \left[ Q(s,a) + c\sqrt{2\frac{\ln T_s}{T_{s,a}}} \right], \tag{3}$$

where $Q(s,a)$ is the estimated value of action $a$ at $s$, $T_s$ is the visit count of $s$, $T_{s,a}$ is the visit count of $(s,a)$, and $c$ controls exploration. Unvisited actions ($T_{s,a} = 0$) yield infinite scores, ensuring exploration.

In the *expansion* phase, a child node is added for an unvisited action. The *simulation* phase then performs a rollout from this node to a terminal state. In the *backpropagation* phase, the obtained reward is propagated along the visited path, updating statistics.

This process repeats until the computational budget is exhausted, progressively improving the search estimates.

## 3 Methodology

In this section, we outline two improvements to the Monte Carlo Tree Search for symbolic regression: the allocation strategy based on the extreme bandit and the incorporation of the state-jumping actions.

### 3.1 Extreme Bandit Allocation Strategy

The objective of symbolic regression is to identify the optimal expression, which corresponds to the highest reward, rather than a policy that maximizes the expected cumulative rewards. Consequently, the UCB allocation strategy (3), which is optimal for maximizing the expected cumulative rewards, may not be well-suited for this task. This limitation has been empirically demonstrated in [16, 27–29]. In this subsection, we introduce an extreme bandit allocation strategy, similar to UCB, with optimal guarantees under certain reward distribution assumptions.

The selection step at state $s$ is modeled as a $K$ arm extreme bandit problem [33–35], where each action corresponds to an arm. We assume that choosing arm $k$ at time $t$ yields a reward $X_{k,t}$, determined through the subsequent selection until reaching a leaf node, followed by the simulation step. We assume the reward is drawn from an unknown distribution $P_k$ over $[0, 1]$ and $P_k$ is supported on $[0, b_k]$ with $b_k \leq 1$. Discovering the best expression is equivalent to discovering the maximum reward $\max_{k=1}^{K}\{b_k\}$. Given a designed allocation strategy, suppose arm $I_t$ is selected at time $t$. The objective is to design an allocation strategy that minimizes

$$J(T) = \max_{k=1}^{K}\{b_k\} - \mathbb{E}\Big[\max_{t \leq T} X_{I_t,t}\Big], \tag{4}$$

where the second term represents the expected highest reward achieved under our strategy. The objective function (4) can be decomposed into two components: The performance gap, which quantifies the difference between the maximum possible reward and the expected highest reward when the optimal action is known, accounting for the randomness of the reward:

$$G(T) = \max_{k=1}^{K}\{b_k\} - \max_{k=1}^{K}\Big\{\mathbb{E}\Big[\max_{t \leq T} X_{k,t}\Big]\Big\}. \tag{5}$$

And the regret, as defined in [35]

$$R(T) = \max_{k=1}^{K}\Big\{\mathbb{E}\Big[\max_{t \leq T} X_{k,t}\Big]\Big\} - \mathbb{E}\Big[\max_{t \leq T} X_{I_t,t}\Big]. \tag{6}$$

Therefore, determining the optimal allocation strategy is equivalent to minimizing the regret. The term $G(T)$ is independent of our strategy and serves as a fundamental performance limit.

In this work, we consider the following allocation strategy for the $T + 1$-th round:

$$I_{T+1} := \arg\max_k\Big\{\hat{Q}_{k,T_k} + 2c\big(\frac{\ln T}{T_{k,T}}\big)^\gamma\Big\} \qquad \text{with} \qquad \hat{Q}_{k,T_{k,T}} = \max_{t:I_t=k} X_{I_t,t}, \tag{7}$$

where $T_{k,T}$ denotes the number of times the $k$-th arm has been selected, and $\hat{Q}_{k,T_k}$ denotes the highest observed reward obtained from the $k$-th arm so far. The parameters $c > 0$ and $\gamma > 0$ are two hyperparameters satisfying condition in (10). We now establish upper bounds on the performance gap and the regret, assuming that the reward distribution of each arm exhibits polynomial decay near its maximum:

**Theorem 1** (Polynomial-like Arms Upper Bounds). *Assume there are $K$ arms, where the rewards for each arm follow a special beta distribution $X_{k,t} \sim P(x; a_k, b_k)$ supported on $[0, b_k]$:*

$$P(x; a, b) = 1 - (1 - \frac{x}{b})^a \quad \text{with} \quad a \geq 1 \text{ and } b \leq 1. \tag{8}$$

*We further assume that the first arm is the optimal, meaning $\Delta_k = b_1 - b_k > 0, \forall k \geq 2$. Then the performance gap satisfies*

$$G(T) \leq \frac{b_1}{(T + \frac{1}{a_1})^{\frac{1}{a_1}}}. \tag{9}$$

*Furthermore, we consider the allocation strategy defined in* (7) *with*

$$\frac{1}{\gamma} \geq a_1 \qquad \text{and} \qquad 2^{a_1} c^{\frac{1}{\gamma}} \geq 2 + \frac{1}{a_1}. \tag{10}$$

*Then for $T \geq C \ln T + K$ with constant*

$$C = \sum_{k=2}^{K}\Big(\frac{2c}{\Delta_k}\Big)^{1/\gamma}, \tag{11}$$

*the regret bound is given by*

$$R(T) \leq K^2 b_1 \frac{2^{a_1} c^{\frac{1}{\gamma}} - 1}{2^{a_1} c^{\frac{1}{\gamma}} - 2} \frac{C \ln T + 2K}{(T - C \ln T - K)^{1 + \frac{1}{a_1}}}. \tag{12}$$

The proof of Theorem 1 is provided in Appendix A.1. In the reward density assumption (8), the parameter $a$ controls the decay rate of the reward density near the maximum $b$. Larger values of $a$ indicate that fewer expressions can achieve high rewards. The theorem establishes that

$$G(T) = \mathcal{O}(\frac{1}{T^{1/a_1}}) \quad \text{and} \quad R(T) = \mathcal{O}(\frac{\ln T}{T^{1+1/a_1}}),$$

which depends only on the tail decay rate $a_1$ of the optimal arm. When $a_1$ is large, the performance gap increases; however, the regret remains at least $\mathcal{O}(\frac{\ln T}{T})$. Together with the following theorem, this result demonstrates that the UCB-extreme allocation strategy (7) is also optimal for such extreme bandit problems. However, the required hyperparameters must satisfy (10). A key condition is that $\frac{1}{\gamma}$ must be greater than the polynomial decay rate, which is generally unknown and varies case by case. While choosing a small $\gamma$ satisfies this condition, it also causes the constant $C$ defined in (11) in the regret bound to grow exponentially with $\frac{1}{\gamma}$. In practice, $a_1$ can be estimated, and one may, for example, set $\gamma = \frac{1}{a_1}$ and $c = \frac{1}{2} + \frac{1}{a_1}$, which satisfy the condition (10). The estimation of $a_1$ for the symbolic regression task is discussed in Section 4 and Appendix F. Meanwhile, we have also conducted corresponding numerical experiments to verify this theorem and demonstrate the effectiveness of the UCB-extreme strategy, as detailed in Appendix C.

**Theorem 2** (Polynomial-like Arms Lower Bounds). *Assume there are $K$ arms, where the rewards for each arm follow a special beta distribution $X_{k,t} \sim P(x; a_k, b_k)$ supported on $[0, b_k]$:*

$$P(x; a, b) = 1 - (1 - \frac{x}{b})^a \quad \text{with} \quad a \geq 1 \text{ and } b \leq 1.$$

*We further assume that the first arm is the optimal, meaning $\Delta_k = b_1 - b_k > 0, \forall k \geq 2$, and $b_1 < 1$. Consider a strategy that satisfies $\mathbb{E}[T_{k,T}] = o(T^\delta)$ as $T \to \infty$ for any arm $k$ with $\Delta_k > 0$, and any $\delta > 0$. Then, the following holds*

$$\lim \inf_{T \to \infty} \mathbb{E}[T_{k,T}] \geq \frac{\ln T}{\mathrm{KL}[P(x; a_k, b_k) \| P(x; a_1, b_1)]} \tag{13}$$

*where KL denotes the Kullback–Leibler divergence (relative entropy) between the two distributions.*

The proof of Theorem 2 follows from [44–46], and is presented in Appendix A.2. Finally, we point out that for more challenging reward distributions, which decay faster than polynomials, it becomes difficult (or even impossible) to identify the best expressions. We focus on reward distributions that exhibit exponential decay near their maximum. The following negative result indicates that the performance gap is $\mathcal{O}(\frac{1}{\ln T})$, meaning that an exponentially increasing number of samples is required to find $\max_{k=1}^{K}\{b_k\}$ or to identify the best expression with high probability.

**Theorem 3** (Exponential-like Arms). *Assume there are $K$ arms, where the rewards for each arm follow a distribution $X_{k,t} \sim P(x; a_k, b_k)$ supported on $[0, b_k]$:*

$$P(x; a, b) = 1 - e^{-\frac{ax}{b-x}} \quad \text{with} \quad a > 0 \text{ and } b \leq 1. \tag{14}$$

*We further assume that the first arm is the optimal, meaning $\Delta_k = b_1 - b_k > 0, \forall k \geq 2$. Then the performance gap satisfies*

$$G(T) \geq \min\left\{\frac{a_1 b_1}{e \ln(T+1)}, \min_{k \geq 2} \Delta_k\right\}. \tag{15}$$

The proof of Theorem 3 is presented in Appendix B.

## 3.2 Evolution-inspired State-jumping Actions

To improve search efficiency, we integrate mutation and crossover from GP as state-jumping actions within MCTS. Unlike prior hybrid methods [23, 24, 29] which alternate between GP and RL, our framework tightly embeds these actions into MCTS. As illustrated in Figure 1, each MCTS node maintains a priority queue of high-reward expressions. During selection, we probabilistically trigger state-jumping, applying mutation or crossover using expressions from this queue. These jumps guide the search toward high-reward regions by leveraging past successful expressions, effectively bypassing less promising paths. To ensure consistency and maximize the utility of information within

**Algorithm 1:** Improved MCTS
___
**Input:** Initial state $s_0$, $p_s$: state-jumping rate, $p_m$: mutation rate, $\epsilon$: random explore rate
**Output:** Updated MCTS tree
1  Initialize $v \leftarrow v_0$, $s \leftarrow s_0$, $T_v \leftarrow T_v + 1$                                          ▷ Init
2  **while** NOTLEAF(v) **do**                           ▷ Until reaching a leaf node
3     Let $\xi \sim$ UNIFORM$(0, 1)$                        ▷ Random variable
4     **if** $\xi_1 < p_s$:                         ▷ Evolutionary state-jumping
5       **if** $\xi_2 < p_m$:
6         $\tau_m \leftarrow$ MUT$(s, v)$, BACK/FORWARDprop(v, $\tau_m$)          ▷ Mutation
7       **else**:
8         **for** $\tau_c$ in CRS(s,v): BACK/FORWARDprop(v, $\tau_c$)       ▷ Crossover
9     **if** $\xi_3 < \epsilon$: $v \leftarrow \mathcal{U}(\mathcal{C}(v))$                 ▷Randomly explore
10    **else**: $v \leftarrow \arg\max_{v'} \Psi(v')$              ▷UCB-extreme exploit
11    $s \leftarrow f(s, a_v)$, $T_v \leftarrow T_v + 1$              ▷ State transition
12  **end while**
13  **if** NONTERMINAL(s): $v \leftarrow$ EXPAND$(v)$, update $s$
14  BACKPROP$(v,$ SIMULATE$(s))$               ▷Simulation and update ancestral nodes
15  **for** $\tau_p \in \mathcal{Q}_{v^p}$: FORWARDPROP$(v^p, \tau_p)$      ▷ Propagate parent node's results to child nodes
___

the MCTS tree, we introduce bidirectional propagation, which updates the priority queues of both ancestors and descendants whenever a high-reward expression is found. This enables efficient sharing of valuable expressions across the tree and enhances overall search performance.

**Priority Queue.** For each MCTS node $v$, we maintain a priority queue $\mathcal{Q}(v)$ that stores the Top-$N$ reward-trajectories, each denoted as $\tau = (a_h, a_{h+1}, \cdots)$ along with their associated rewards $r(\tau)$. Let $s$ denote the state at $v$, so that the sequence $(s, a_h, a_{h+1}, \cdots)$ defines a complete symbolic expression, with the reward given by $r(\tau)$. The queue is dynamically updated, recording the top-$N$ highest reward trajectories from both standard MCTS iterations and state-jumping actions passing through the node, even before this MCTS node is expanded. This is achieved through the following bidirectional propagation mechanism.

**Bidirectional Propagation.** Conventional MCTS (Section 2.3) updates node information through backpropagation (upward propagation to the root), which includes the node visit count and the $Q$-value. However, backpropagation alone is insufficient for maintaining the priority queue and propagating information for state-jumping actions. For instance, if a node is visited during the simulation step but has not yet been expanded, the information for the top-$N$ reward trajectories and their associated rewards is not recorded in that node. To address this, we perform downward propagation after MCTS node expansion. Specifically, when a new MCTS node is expanded, its parent node downward propagates information, including the trajectories that bypass the newly expanded node and the associated rewards. Additionally, after a state-jumping action, information is propagated both upward and downward. With this bidirectional propagation, for any node $v$ with state $s$, the highest reward stored in the priority queue $\mathcal{Q}(v)$, denoted as $\hat{V}(v)$, equals the maximum reward across all complete trajectories passing through $v$. Furthermore, this highest reward is also the maximum among its child nodes' highest rewards

$$\hat{V}(s) = \max\{\hat{V}(v') : v' \text{ is a child node of } v\}.$$

More generally, the top-$N$ reward queue of the parent node $\mathcal{Q}(v)$ collects the top-$N$ trajectories from the reward queues of its child nodes. Additionally, if the edge connecting $v$ and its child $v'$ corresponds to action $a$, then the highest reward satisfies $\hat{V}(v') = \hat{Q}(s, a)$. Implementation details are provided in Appendix G.

**State-jumping Actions.** We integrate evolution-inspired state-jumping actions, including crossover and various mutation types—following the implementations in [23, 29, 47], into the MCTS process. During each iteration, a state-jumping action is applied at a node $v$ with depth-dependent probability $p_s$, which decreases exponentially with depth. This design prioritizes high-reward nodes near the root, whose top-$N$ queues typically contain better candidates. With probability $\epsilon$, node selection deviates from the extreme bandit allocation strategy (7), choosing nodes uniformly at random before applying a state-jumping action. These actions are exclusive to the MCTS phase and do not appear in rollouts.

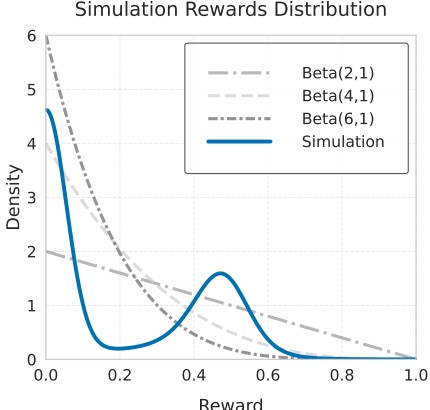 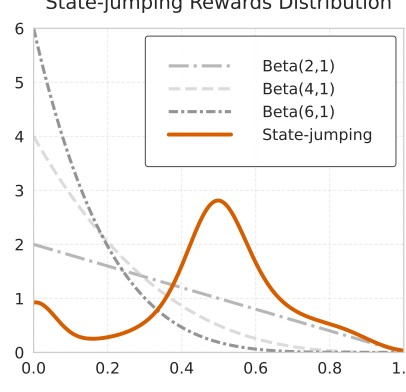

Figure 2: Empirical reward distribution for Nguyen-4 under $\epsilon = 1$ in Algorithm 1 (fully random node selection), with a computational budget of 200,000 expression evaluations. All other settings follow Table 4. Left: standard MCTS simulation. Right: state-jumping actions. A Gaussian kernel density estimate (bandwidth 0.25) is computed over 100 runs. Overlaid gray curves represent beta distributions on $[0, 1]$ with varying tail decay parameters $a = 2, 4, 6$.

The newly generated states and their rewards are incorporated via the bidirectional propagation mechanism.

State-jumping actions extend the search beyond sequential symbol updates by enabling direct transitions to distant states, thus enhancing global exploration. Meanwhile, the bidirectional propagation mechanism retains and shares useful substructures across the tree. Combined with the extreme bandit allocation strategy, this hybrid approach improves the reward distribution in MCTS, which improves robustness to hyperparameters and facilitates the generation of more complex expressions, as discussed in Section 4. Detailed pseudocode of the algorithm is provided in Algorithm 1.

## 4 Experiment and Result

In this section, we assess our method on a variety of benchmarks. The **Basic Benchmarks** include several ground-truth datasets where the true closed-form expressions are known: Nguyen [16], Nguyen$^C$ [16], Jin [48], and Livermore [23]. These datasets consist of data points sampled from equations with at most two variables over restricted intervals. Notably, here we replace Nguyen-12 with Nguyen-12*[23]. The **SRBench Black-box Benchmarks (SRBench)** [10, 49] feature more challenging datasets: Feynman [17], Strogatz [50], and the Black-box collection. The Black-box subset contains 122 tasks with two or more input variables: 46 are drawn from real-world observational datasets, and 76 are synthetic problems derived from analytical functions or simulation models. Detailed experimental parameters and procedures are provided in Appendix H. For benchmarks involving constants, each evaluation of a symbolic expression based on (2) requires optimizing the constants within the expression using the BFGS algorithm [51] as implemented in `SciPy` [52].

**Algorithm Analysis.** The effects of the extreme bandit allocation strategy and the state-jumping actions introduced in Section 3 are analyzed using the Nguyen benchmarks. We first visualize the reward distribution—an important indicator of high-probability recovery, as discussed in Section 3.1. Most Nguyen test cases exhibit polynomial tail decay rates approximately in $[4, 6]$, a representative example for Nguyen-4 is presented in Figure 2. Next, we test MCTS only equipped with the UCB-extreme strategy, under different parameter configurations (10). The results show that, for an estimated tail decay rate above $a_1 = 6$, the algorithm can achieve optimal recovery performance; however, the algorithm is highly sensitive to $c$ and requires careful tuning to maintain robust performance across tasks. Moreover, it often struggles on problems with more complex target expressions (e.g., Nguyen-4). Finally, we examine the effect of incorporating state-jumping actions, which dramatically reshape the reward landscape. Before adding state-jumps, the estimated reward tail decay—measured by the $a$ (first) parameter of the Beta distribution—fell in the interval $[4, 6]$ for Nguyen-4 (see Figure 2). After introducing state-jumping, that same $a$ value drops below 2. Across all other Nguyen

Table 1: Average recovery rate (%) of original expressions on five ground-truth benchmarks. The "Datasets" column indicates the number of individual datasets per benchmark. Results are averaged over 100 runs under a 2 million-evaluation budget.

| Benchmark | Datasets | Ours | DSR | GEGL | NGGP | PySR |
|-----------|----------|------|-----|------|------|------|
| Nguyen | 12 | **93.25** | 83.58 | 86.00 | 92.33 | 74.41 |
| Nguyen$^C$ | 5 | **100.00** | **100.00** | **100.00** | **100.00** | 65.40 |
| Jin | 6 | **100.00** | 70.33 | 95.67 | **100.00** | 72.17 |
| Livermore | 22 | **71.41** | 30.41 | 56.36 | 71.09 | 46.14 |

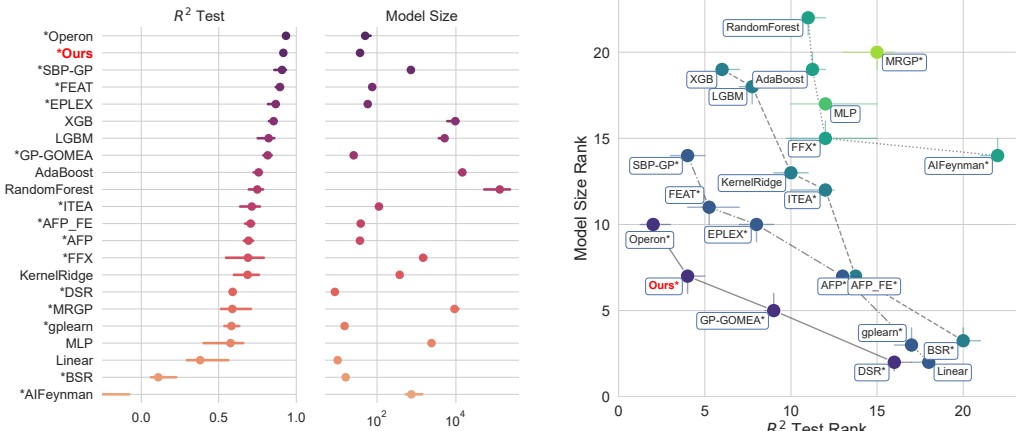

Figure 3: Comparison of our algorithm and SRBench baselines on the Black-box benchmark, showing median test $R^2$ and model size across 122 problems (95% confidence intervals; asterisks denote symbolic regression methods); and the Pareto frontier of model size vs. median test $R^2$ (median rankings, 95% confidence intervals; lines/colors indicate Pareto dominance).

test cases, the fitted $a$ shifts from its original range into the interval $[2, 4]$. In practical terms, lowering $a$ in this way leads to faster and more stable convergence. Additional decay rate estimates for the full Nguyen suite can be found in Appendix F. Accordingly, in the subsequent comparative experiments we fix $\frac{1}{\gamma} = 2, c = 1$. Additionally, we also performed ablation experiments on Nguyen benchmark, which can be found in Appendix E.

**Comparison Study.** We benchmark our algorithm (Algorithm 1) against several representative baselines. Table 1 reports the average recovery rate on the Basic Benchmarks over 100 independent runs, using a budget of 2 million expression evaluations. The baselines include the RNN-based RL algorithm DSR [16], GEGL [53], NGGP [23], and the genetic programming library PySR [15]. GEGL combines genetic programming with imitation learning, while NGGP integrates it with reinforcement learning following DSR. These methods were selected as they are general-purpose algorithms suitable for combinatorial optimization, similar to ours. Our approach achieves comparable performance to these baselines.

We further evaluate our algorithm on SRBench, comparing it with 21 baseline methods reported in [10]. As shown in Figure 3, our method strikes a favorable balance between accuracy and model complexity: it ranks second in test accuracy, just behind Operon [13], while producing substantially simpler models. On the Pareto frontier, our method stands among the top-performing approaches, alongside Operon [13], GP-COMEA [14], and DSR [16].

## 5 Discussion and Limitation

We propose an improved MCTS framework for symbolic regression with two key innovations: (1) an extreme bandit allocation strategy, and (2) evolution-inspired state-jumping actions. The extreme

bandit strategy is based on best-arm identification with polynomial-like reward and offers theoretical guarantees for optimal finite-time regret. Meanwhile, the state-jumping actions reshape the reward landscape towards high-reward regions. The proposed method achieves competitive performance on various symbolic regression benchmarks, including ground-truth and complex SRBench black-box tasks. Despite these advances, some limitations remain. First, the bandit strategy's theoretical guarantees depend on reward distribution assumptions, which future work could explore relaxing. Second, certain challenging problems, such as Nguyen-12, remain unsolved. Finally, extending these innovations to broader reinforcement learning and combinatorial optimization domains is a promising future direction.

## Acknowledgement

The authors acknowledge the support of the National Natural Science Foundation of China through grant 12471403, the Fundamental Research Funds for the Central Universities, and the high-performance computing platform of Peking University.

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

# A  Polynomial-like Arms

In this section, we study the extreme bandit problem with Polynomial-like arms, assuming that the reward distribution of each arm exhibits polynomial decay near its maximum $b$:

$$P(X > x; a, b) \sim (1 - \frac{x}{b})^a \quad \text{with} \quad a \geq 1 \text{ and } b \leq 1.$$

Specifically, we consider beta distribution. We have the following preliminary about beta distribution

**Lemma 1.** *For beta distribution with cumulative density function*

$$P(x; a, b) = 1 - (1 - \frac{x}{b})^a \quad \text{with} \quad a \geq 1 \text{ and } b \leq 1.$$

*The probability density function (PDF) of its maximum over $n$ samples is $\rho_n$, its expectation satisfies*

$$\mathbb{E}_{\rho_n}[x] = b\Big(1 - F(a, n)\Big) \quad \text{where} \quad F(a, n) = \Gamma(\frac{1}{a} + 1)\frac{\Gamma(n + 1)}{\Gamma(\frac{1}{a} + n + 1)}$$

*And $F(a, n)$ satisfies*

$$\Gamma(\frac{1}{a} + 1)(n + \frac{1}{a} + 1)^{-\frac{1}{a}} \leq F(a, n) \leq \Gamma(\frac{1}{a} + 1)(n + \frac{1}{a})^{-\frac{1}{a}} \tag{16}$$

$$F(a, n_1) - F(a, n) \leq \Gamma(\frac{1}{a} + 1)\frac{n - n_1 + 1}{(n_1 + \frac{1}{a})(n + \frac{1}{a})^{\frac{1}{a}}} \quad (n_1 \leq n) \tag{17}$$

*The density is highly concentrated near its maximum, b, with*

$$P_n\Big(x + \epsilon \leq b\Big) = \Big(1 - (\frac{\epsilon}{b})^a\Big)^n \leq e^{-n(\frac{\epsilon}{b})^a} \tag{18}$$

*Proof.* For the beta distribution $P(x; a, b)$, the probability density function (PDF) is given by:

$$\rho(x; a, b) = \frac{a}{b}(1 - \frac{x}{b})^{a-1} \ (0 \leq x \leq b).$$

The PDF of its maximum over $n$ samples is

$$\rho(x) = \frac{na}{b}(1 - \frac{x}{b})^{a-1}\Big(1 - (1 - \frac{x}{b})^a\Big)^{n-1} \ (0 \leq x \leq b)$$

with expectation

$$\mathbb{E}_{\rho_n}[x] = n\int_0^b x\frac{a}{b}(1 - \frac{x}{b})^{a-1}\Big(1 - (1 - \frac{x}{b})^a\Big)^{n-1}dx$$

$$= nb\int_0^1 (1 - (1 - y)^{\frac{1}{a}})y^{n-1}dy \qquad (y = 1 - (1 - \frac{x}{b})^a) \tag{19}$$

$$= b\Big(1 - n\frac{\Gamma(\frac{1}{a} + 1)\Gamma(n)}{\Gamma(\frac{1}{a} + n + 1)}\Big)$$

Let denote

$$F(a, n) = \Gamma(\frac{1}{a} + 1)\frac{\Gamma(n + 1)}{\Gamma(\frac{1}{a} + n + 1)},$$

we have

$$\mathbb{E}_{\rho_n}[x] = b(1 - F(a, n)).$$

By using the following inequality [54] about $\Gamma$ function

$$(n + \frac{1}{a} + 1)^{-\frac{1}{a}} \leq \frac{\Gamma(n + 1)}{\Gamma(\frac{1}{a} + n + 1)} \leq (n + \frac{1}{a})^{-\frac{1}{a}} \tag{20}$$

We have (16). For $n_1 \leq n$, we have

$$F(a, n_1) - F(a, n) \leq \Gamma(\frac{1}{a} + 1)\Big((n_1 + \frac{1}{a} + 1)^{-\frac{1}{a}} - (n + \frac{1}{a})^{-\frac{1}{a}}\Big) \leq \Gamma(\frac{1}{a} + 1)\frac{n - n_1 + 1}{(n_1 + \frac{1}{a})(n + \frac{1}{a})^{\frac{1}{a}}} \tag{21}$$

$\square$

### A.1 Upper Bounds

For the extreme bandit problem, we have the following theorem about the performance gap.

**Theorem 4.** *Assume there are $K$ arms, where the rewards for each arm follow a distribution $X_{k,t} \sim P(x; a_k, b_k)$ supported on $[0, b_k]$:*

$$P(x; a, b) = 1 - (1 - \frac{x}{b})^a \quad with \quad a \geq 1 \ and \ b \leq 1.$$

*We further assume that the first arm is the optimal, meaning $\Delta_k = b_1 - b_k > 0, \forall k \geq 2$. Then the performance gap satisfies*

$$G(T) \leq \frac{b_1}{(T + \frac{1}{a_1})^{\frac{1}{a_1}}}$$

*Proof.* By using (16), we have

$$G(T) = b_1 - \max_{k=1}^{K}\left\{\mathbb{E}_{\rho_T(x; a_k, b_k)}[x]\right\} \leq b_1 - \mathbb{E}_{\rho_T(x; a_1, b_1)}[x] = b_1 F(a_1, T) \leq \frac{b_1 \Gamma(\frac{1}{a_1} + 1)}{(T + \frac{1}{a})^{-\frac{1}{a}}}$$

Using the fact that $\Gamma(\frac{1}{a_1} + 1) \leq 1$ leads to bound for the performance gap. $\square$

Finally, we will prove Theorem 1 about the regret bound related to our allocation strategy (7).

**Theorem 5.** *Assume there are $K$ arms, where the rewards for each arm follow a distribution $X_{k,t} \sim P(x; a_k, b_k)$ supported on $[0, b_k]$:*

$$P(x; a, b) = 1 - (1 - \frac{x}{b})^a \quad with \quad a \geq 1 \ and \ b \leq 1.$$

*We further assume that the first arm is the optimal, meaning $\Delta_k = b_1 - b_k > 0, \forall k \geq 2$, and denote*

$$C = \sum_{k=2}^{K} \left(\frac{2c}{\Delta_k}\right)^{1/\gamma}. \tag{22}$$

*We consider the allocation strategy*

$$I_{T+1} := arg \, max_k \left\{\hat{Q}_{k,T_k} + 2c\left(\frac{\ln T}{T_{k,T}}\right)^\gamma\right\} \qquad with \qquad \hat{Q}_{k,T_k,T} = \max_{t: I_t = k} X_{I_t, t} \tag{23}$$

*with $\frac{1}{\gamma} \geq a_1$ and $2^{a_1} c^{\frac{1}{\gamma}} \geq 2 + \frac{1}{a_1}$. Then for $T \geq C \ln T + K$, the regret bound is given by*

$$R(T) \leq K^2 b_1 \frac{2^{a_1} c^{\frac{1}{\gamma}} - 1}{2^{a_1} c^{\frac{1}{\gamma}} - 2} \frac{C \ln T + 2K}{(T - C \ln T - K)^{1+a_1}} \tag{24}$$

*Proof.* We will first prove that under the conditions

$$\mathbb{E}\left[\max_{t \leq T} X_{1,t}\right] \geq \mathbb{E}\left[\max_{t \leq T} X_{k,t}\right] \tag{25}$$

which is equivalent to

$$
\begin{aligned}
b_1 - b_1 F(a_1, T) - b_k + b_k F(a_k, T) &\geq \Delta_k - b_1\left(T + \frac{1}{a_1}\right)^{-\frac{1}{a_1}} && \text{using (16) and } \Gamma(\frac{1}{a_1} + 1) \leq 1 \\
&\geq \Delta_k - b_1\left(\frac{2c}{\Delta_k}\right)^{-\frac{1}{\gamma a_1}} && \text{using } T \geq C \ln T + K \geq \left(\frac{2c}{\Delta_k}\right)^{1/\gamma} \\
&= \Delta_k - \Delta_k b_1 \frac{\Delta_k^{\frac{1}{\gamma a_1} - 1}}{(2c)^{\frac{1}{\gamma a_1}}} \\
&\geq 0
\end{aligned}
\tag{26}
$$

In the last inequality, we used $\frac{1}{\gamma} \geq a_1$ and $2c \geq 1$, which is derived from $\frac{(2c)^{\frac{1}{\gamma}}}{2^{\frac{1}{\gamma}-a_1}} = 2^{a_1} c^{\frac{1}{\gamma}} \geq 2 + \frac{1}{a_1}$.

Then the regret for the first $T$ round is upper bounded by

$$R(T) \leq b_1 \Big( \mathbb{E}\Big[ F(a_1, T_{1,T}) \Big] - F(a_1, T) \Big) \tag{27}$$

Here $T_{k,T}$ denotes the number of rounds in which arm $k$ is chosen, and we used the fact that the extreme reward over all rounds is larger than the extreme reward within each individual arm:

$$\mathbb{E}\Big[ \max_{t \leq T} X_{I_t, t} \Big] \geq \max_{k=1}^{K} \mathbb{E}\Big[ \max_{t \leq T, I_t = k} X_{I_t, t} \Big]$$

Our goal is to establish an upper bound on the regret in (27).

Assume that at the $t+1$-th round, when arm $k \neq 1$ is pulled, the allocation strategy given by (7) implies the following inequality

$$1 + 2c\Big(\frac{\ln t}{T_{k,t}}\Big)^{\gamma} \geq \hat{Q}_{k,T_{k,t}} + 2c\Big(\frac{\ln t}{T_{k,t}}\Big)^{\gamma} \geq \hat{Q}_{1,T_{1,t}} + 2c\Big(\frac{\ln t}{T_{1,t}}\Big)^{\gamma} \geq 2c\Big(\frac{\ln t}{T_{1,t}}\Big)^{\gamma}. \tag{28}$$

We define the event $A_t : \hat{Q}_{1,T_{1,t}} + 2c\big(\frac{\ln t}{T_{k,t}}\big)^{\gamma} \leq b_1$. The probability of this event satisfies

$$P(A_t) = \Big(1 - \Big(\frac{2c}{b_1}\big(\frac{\ln t}{T_{k,t}}\big)^{\gamma}\Big)^{a_1}\Big)^{T_{1,t}} \leq e^{-(\frac{2c}{b_1})^{a_1}(\ln t)^{\gamma a_1} T_{1,t}^{1-\gamma a_1}}. \tag{29}$$

Here, we used the fact that $\hat{Q}_{1,T_{1,t}}$ is the extreme value, as described in (18). When the event $A_t$ does not occur, the allocation strategy implies

$$b_k + 2c\Big(\frac{\ln t}{T_{k,t}}\Big)^{\gamma} \geq \hat{Q}_{k,T_{k,t}} + 2c\Big(\frac{\ln t}{T_{k,t}}\Big)^{\gamma} \geq \hat{Q}_{1,T_{1,t}} + 2c\Big(\frac{\ln t}{T_{k,t}}\Big)^{\gamma} \geq b_1,$$

which leads to

$$T_{k,t} \leq \Big(\frac{2c}{b_1 - b_k}\Big)^{1/\gamma} \ln t. \tag{30}$$

When (30) is violated, that indicates that $A_t$ occurs, and the probability is at most as given in (29), which is

$$P(I_{t+1} = k) \leq e^{-(\frac{2c}{b_1})^{a_1}(\ln t)^{\gamma a_1} T_{1,t}^{1-\gamma a_1}}. \tag{31}$$

Then, we begin estimating the regret from (27). We denote the event $B_T$ as

$$B_T : \sum_{k=2}^{K} T_{k,T} \leq C \ln T + K - 1 \qquad \text{with} \qquad C = \sum_{k=2}^{K} \Big(\frac{2c}{b_1 - b_k}\Big)^{1/\gamma}. \tag{32}$$

And we decompose its complement as $B_T^c = \cup_{k=2}^{K} B_k$, where

$$B_k : \Big\{ k = \arg\max_{k=2}^{K}\Big\{ T_{k,T} - \Big(\frac{2c}{b_1 - b_k}\Big)^{1/\gamma} \ln T \Big\} \quad \text{and} \quad \sum_{k=2}^{K} T_{k,T} > C \ln T + K - 1 \Big\} \tag{33}$$

Using the definition of $C$, under event $B_k$, we have

$$1 < T_{k,T} - \Big(\frac{2c}{b_1 - b_k}\Big)^{1/\gamma} \ln T \tag{34}$$

and for all $j > 1$:

$$T_{j,T} - \Big(\frac{2c}{b_1 - b_j}\Big)^{1/\gamma} \ln T \leq T_{k,T} - \Big(\frac{2c}{b_1 - b_k}\Big)^{1/\gamma} \ln T < T_{k,T} - 1. \tag{35}$$

Here we used $\Big(\frac{2c}{b_1 - b_k}\Big)^{1/\gamma} \ln T > (2c)^{1/\gamma} \ln 2 > 1$. We can now estimate the regret in (27) using the following decomposition:

$$\mathbb{E}\Big[ F(a_1, T_{1,T}) - F(a_1, T) \Big] \leq \mathbb{E}\Big[ F(a_1, T_{1,T}) - F(a_1, T) | B_T \Big] P(B_T) \tag{36}$$

$$+ \sum_{k=2}^{K} \mathbb{E}\Big[ F(a_1, T_{1,T}) - F(a_1, T) | B_k \Big] P(B_k) \tag{37}$$

The intuition for the decomposition is as follows: in the first term, $T_{1,T}$ is large enough, and hence $F(a_1, T_{1,T}) - F(a_1, T)$ can be bounded using (16). For the second term, $P(B_k)$ is small due to (34) violates (30). Next, we will estimate each term in the decomposition.

For the first term in (36), under $B_T$, we have $T_{1,T} = T - \sum_{k=2}^{K} T_{k,T} \geq T - C \ln T - K + 1$. By using (17), the first term satisfies

$$\mathbb{E}[F(a_1, T_{1,T}) - F(a_1, T)|B_T]P(B_T) \leq \Gamma(\frac{1}{a_1} + 1)\frac{C \ln T + K}{(T - C \ln T - K + 1 + \frac{1}{a_1})(T + \frac{1}{a_1})^{\frac{1}{a_1}}} \tag{38}$$

For the second term in (36), we further decompose each $B_k$ $(2 \leq k)$ based on the last round number that arm $k$ is chosen. Under event $B_k$, let $t + 1$ denote the last round that arm $k$ is chosen, we have

$$T_{k,t} = T_{k,T} - 1 > \left(\frac{2c}{b_1 - b_j}\right)^{1/\gamma} \ln T > \left(\frac{2c}{b_1 - b_j}\right)^{1/\gamma} \ln t \tag{39}$$

here we used (34) and $t + 1 \leq T$, which violates (30). Hence (31) holds and the probability is at most

$$P(B_k, t) \leq e^{-(\frac{2c}{b_1})^{a_1}(\ln t)^{\gamma a_1}T_{1,t}^{1-\gamma a_1}} \leq t^{-\frac{(2c)^{\frac{1}{\gamma}}}{b_1^{a_1}(1+b_1-b_k)^{\frac{1}{\gamma}-a_1}}} \leq t^{-2^{a_1}c^{\frac{1}{\gamma}}} \tag{40}$$

Here the second inequality is derived from $\gamma \leq \frac{1}{a_1}$ and

$$T_{1,t} \geq \left(\frac{2c}{1 + b_1 - b_j}\right)^{1/\gamma} \ln t,$$

which is obtained by replacing $T_{k,t}$ in (28) as its lower bound in (39). The third inequality uses the fact that $b_1 \leq 1$. Then each term in the second term in (36) can be decomposed as

$$\mathbb{E}[F(a_1, T_{1,T}) - F(a_1, T)|B_k]P(B_k) = \sum_{t=1}^{T-1} \mathbb{E}[F(a_1, T_{1,T}) - F(a_1, T)|B_k, t]P(B_k, t). \tag{41}$$

By combining $T_{k,t} > 0$ from (39) with the condition $t \geq T_{k,t}$, we set $t$ to start from 1.

Then we divide the range of $t$ in (41) into two parts. When $t \geq \frac{T - C \ln T}{K}$, using (40) leads to

$$\sum_{t \geq \frac{T-C \ln T}{K}}^{T-1} \mathbb{E}[F(a_1, T_{1,T}) - F(a_1, T)|B_j, t]P(B_j, t) \leq \sum_{t \geq \frac{T-C \ln T}{K}}^{T-1} t^{-2^{a_1}c^{\frac{1}{\gamma}}}$$
$$\leq \frac{2^{a_1}c^{\frac{1}{\gamma}}}{2^{a_1}c^{\frac{1}{\gamma}} - 1}\left(\frac{K}{T - C \ln T}\right)^{2^{a_1}c^{\frac{1}{\gamma}}-1} \tag{42}$$

Here we used

$$\sum_{t=t_0}^{t_1} t^{-p} \leq t_0^{-p} + \int_{t=t_0}^{\infty} t^{-p}dt = \frac{p}{p-1}t_0^{-p+1} \qquad \forall p > 1. \tag{43}$$

Combining the upper bound of $T_{j,T}$ in (35), and $t \geq T_{k,t} = T_{k,T} - 1$, we have

$$T_{1,T} = T - \sum_{j=2}^{K} T_{j,T} \geq T - C \ln T - (K-1)(T_{k,T} - 1) \geq T - C \ln T - (K-1)t \tag{44}$$

When $t \leq \frac{T - C \ln T}{K}$, using (17) and (40) leads to

$$\sum_{t=1}^{\frac{T-C\ln T}{K}} \mathbb{E}[F(a_1, T_{1,T}) - F(a_1, T)|B_k, t]P(B_k, t)$$

$$\leq \Gamma(\frac{1}{a_1} + 1) \sum_{t=1}^{\frac{T-C\ln T}{K}} \frac{C \ln T + (K-1)t}{(T - C \ln T - (K-1)t + \frac{1}{a_1})(T + \frac{1}{a_1})^{\frac{1}{a_1}}} t^{-2^{a_1} c^{\frac{1}{\gamma}}}$$

$$\leq \Gamma(\frac{1}{a_1} + 1) \sum_{t=1}^{\frac{T-C\ln T}{K}} \frac{C \ln T + (K-1)t}{(\frac{T-C\ln T}{K} + \frac{1}{a_1})(T + \frac{1}{a_1})^{\frac{1}{a_1}}} t^{-2^{a_1} c^{\frac{1}{\gamma}}}$$

$$\leq \Gamma(\frac{1}{a_1} + 1) \frac{KC \ln T + K^2}{(T - C \ln T + \frac{K}{a_1})(T + \frac{1}{a_1})^{\frac{1}{a_1}}} \frac{2^{a_1} c^{\frac{1}{\gamma}} - 1}{2^{a_1} c^{\frac{1}{\gamma}} - 2}$$

(45)

Here in the second inequality, we replaced $t$ in the denominator as $\frac{T - C \ln T - K}{K}$. In the last inequality, we used (43).

Combining (38), (42) and (45), we have the regret bound

$$R(T) \leq b_1 \left( \mathbb{E}\left[ F(a_1, T_{1,T}) - F(a_1, T) \right] \right)$$

$$\leq b_1 \frac{C \ln T + K}{\left(T - C \ln T - K\right)^{1 + \frac{1}{a_1}}}$$

$$+ b_1(K-1) \left( \frac{2^{a_1} c^{\frac{1}{\gamma}}}{2^{a_1} c^{\frac{1}{\gamma}} - 1} \left( \frac{K}{T - C \ln T} \right)^{2^{a_1} c^{\frac{1}{\gamma}} - 1} + \frac{2^{a_1} c^{\frac{1}{\gamma}} - 1}{2^{a_1} c^{\frac{1}{\gamma}} - 2} \frac{KC \ln T + K^2}{\left(T - C \ln T - K\right)^{1 + \frac{1}{a_1}}} \right)$$

$$\leq b_1 \frac{2^{a_1} c^{\frac{1}{\gamma}} - 1}{2^{a_1} c^{\frac{1}{\gamma}} - 2} \frac{K^2 C \ln T + 2K^3}{\left(T - C \ln T - K\right)^{1 + \frac{1}{a_1}}}$$

(46)

Here in the second inequality, we replaced the denominators with the lower bound $\left(T - C \ln T - K\right)^{1 + \frac{1}{a_1}}$ and used the fact that $\Gamma(\frac{1}{a_1} + 1) \leq 1$ for all $a_1 \geq 1$. In the third inequality, we used $T - C \ln T \geq K$, $2^{a_1} c^{\frac{1}{\gamma}} - 1 \geq 1 + \frac{1}{a_1}$ and the following inequality

$$\left( \frac{K}{T - C \ln T} \right)^{2^{a_1} c^{\frac{1}{\gamma}} - 1} \leq \left( \frac{K}{T - C \ln T} \right)^{1 + \frac{1}{a_1}}$$

$\square$

## A.2 Lower Bounds

In this section we will first prove the following theorem, and discuss the optimality of the regret bound.

**Theorem 6** (Polynomial-like Arms Lower Bounds). *Assume there are $K$ arms, where the rewards for each arm follow a distribution $X_{k,t} \sim P(x; a_k, b_k)$ supported on $[0, b_k]$:*

$$P(x; a, b) = 1 - (1 - \frac{x}{b})^a \quad \text{with} \quad a \geq 1 \text{ and } b \leq 1.$$

*We further assume that the first arm is the optimal, meaning $\Delta_k = b_1 - b_k > 0, \forall k \geq 2$, and $b_1 < 1$. Consider a strategy that satisfies $\mathbb{E}[T_{k,T}] = o(T^a)$ as $T \to \infty$ for any arm $k$ with $\Delta_k > 0$, and any $a > 0$. Then, the following holds*

$$\lim \inf_{T \to \infty} \mathbb{E}[T_{k,T}] \geq \frac{\ln T}{KL[P(x; a_k, b_k) \| P(x; a_1, b_1)]}$$

(47)

*where KL denotes the Kullback–Leibler divergence (relative entropy) between the two distributions.*

*Proof.* Without loss of generality, assume that arm 1 is optimal, meaning $b_k < b_1 < 1$ for all $k \geq 2$. Let denote the density function of the beta distribution

$$\rho(x; a_k, b_k) = \frac{a}{b}(1 - \frac{x}{b})^{a-1}.$$

Then the KL divergence between the two reward distributions is given by

$$\text{KL}[P(x; a_k, b_k) \| P(x; a_1, b_1)] = \int \rho(x; a_k, b_k) \ln \frac{\rho(x; a_k, b_k)}{\rho(x; a_1, b_1)} dx$$

This quantity is finite since $b_k < b_1$ and is continuous with respect to $b_1$. Now, given any $\epsilon > 0$, consider an alternative bandit model where $b_k$ is replaced by $b'_k$ such that $b'_k > b_1 > b_k$ and

$$\text{KL}[P(x; a_k, b_k) \| P(x; a_1, b'_k)] \leq (1 + \epsilon) \text{KL}[P(x; a_k, b_k) \| P(x; a_1, b_1)] \tag{48}$$

Under this alternative model, arm $k$ becomes the unique optimal arm. In the following, we show that with big enough probability, the allocation strategy cannot distinguish between the two models, leading to the desired lower bound.

We use the notation $\mathbb{E}'$, and $P'$ to denote expectation and probability under the alternative model where the parameter of arm $k$ is replaced by $b'_k$. For any event $A$ involving the rewards $X_{k,1}, X_{k,2}, \cdots, X_{2,T_{k,T}}$ from arm $k$, the probability under the alternative model satisfies

$$P'(A) = \mathbb{E}[1_A e^{-\widehat{\text{KL}}_{T_{k,T}}}] \qquad \widehat{\text{KL}}_s = \sum_{t=1}^{s} \ln \frac{\rho(X_{k,t}; a_2, b_2)}{\rho(X_{k,t}; a_2, b'_2)} \tag{49}$$

Here, the expectation is with respect to the original model. Define $P_k(x) = P(x; a_k, b_k)$ and $P'_k(x) = P(x; a_k, b'_k)$. Then, we have

$$\lim_{s \to \infty} \frac{\widehat{\text{KL}}_s}{s} \xrightarrow{a.s.} \text{KL}[P_k \| P'_k] = \int \rho(x; a_k, b_k) \ln \frac{\rho(x; a_k, b_k)}{\rho(x; a_1, b'_k)} dx < \infty \tag{50}$$

This follows from the fact that the random variables $\ln \frac{\rho(X_{k,t}; a_k, b_k)}{\rho(X_{k,t}; a_1, b'_k)}$ with $X_{k,t} \sim P_k$ are i.i.d. and have bounded finite moments.

In order to link the behavior of the allocation strategy on the original and the modified bandits we introduce the event

$$A_T = \left\{ T_{k,T} < f_T \quad \text{and} \quad \widehat{\text{KL}}_{T_{k,T}} \leq (1 - \frac{\epsilon}{2}) \ln T \right\} \quad \text{with} \quad f_T = \frac{1 - \epsilon}{\text{KL}(P_k \| P'_k)} \ln T \tag{51}$$

We will first prove that $P(A_T) = o(1)$. Using (49) leads to

$$P'(A_T) = \mathbb{E}[1_{A_T} e^{-\widehat{\text{KL}}_{T_{k,T}}}] \geq T^{-(1 - \frac{\epsilon}{2})} P(A_T) \tag{52}$$

Using Markov's inequality, the above implies

$$P(A_T) \leq T^{1 - \frac{\epsilon}{2}} P'(A_T) \leq T^{1 - \frac{\epsilon}{2}} P'\left(T_{k,T} < f_T\right) \leq T^{1 - \frac{\epsilon}{2}} \frac{\mathbb{E}'[T - T_{k,T}]}{T - f_T} \tag{53}$$

Now note that in the modified model, arm $k$ is the unique optimal arm. By assumption, for any suboptimal arm $j$ and any $\delta > 0$, the strategy satisfies $\mathbb{E}'T_{j,T} = o(T^\delta) \forall j \neq k$. This implies that $\mathbb{E}'[T - T_{k,t}] = o(KT^\delta)$. Choosing $\delta < \epsilon/2$ then leads to

$$P(A_T) \leq T^{1 - \epsilon/2} \frac{\mathbb{E}'[T - T_{k,T}]}{T - f_T} = o(1) \tag{54}$$

Next we will prove $P(T_{k,T} < f_T) = o(1)$. We have

$$P(A_T) \geq P\left(T_{k,T} < f_T \quad \text{and} \quad \max_{s \leq f_T} \widehat{\text{KL}}_s \leq (1 - \frac{\epsilon}{2}) \ln T\right)$$
$$= P\left(T_{k,T} < f_T \quad \text{and} \quad \frac{1}{f_T} \max_{s \leq f_T} \widehat{\text{KL}}_s \leq \frac{1 - \frac{\epsilon}{2}}{1 - \epsilon} \text{KL}[P_k \| P'_k]\right) \tag{55}$$

Here in the first inequality, we introduce maximum operator to eliminate the dependence of KL on $T_{k,T}$. The Using the fact that $\frac{1-\frac{\epsilon}{2}}{1-\epsilon} > 1$ and $\text{KL}[P_k\|P_k'] > 0$, along with (50), the maximal version of the strong law of large numbers [55, Lemma 10.5] implies that

$$\lim_{T\to\infty} P\Big(\frac{1}{f_T}\max_{s\leq f_T}\widehat{\text{KL}}_s \leq \frac{1-\frac{\epsilon}{2}}{1-\epsilon}\text{KL}[P_k\|P_k']\Big) = 1 \tag{56}$$

Combining (56) and (55) leads to that

$$P(T_{k,T} < f_T) = o(1) \tag{57}$$

Finally, we can estimate the expectation of $T_{k,T}$ as follows

$$\begin{aligned}
\mathbb{E}[T_{k,T}] &= \mathbb{E}[T_{k,T}|T_{k,T} < f_T]P(T_{k,T} < f_T) + \mathbb{E}[T_{k,T}|T_{k,T} \geq f_T]P(T_{k,T} \geq f_T) \\
&\geq \frac{1-\epsilon}{\text{KL}[P_2\|P_2']}\ln T\, P(T_{k,T} \geq f_T) \\
&\geq (1+o(1))\frac{1-\epsilon}{1+\epsilon}\frac{\ln T}{\text{KL}[P_2\|P_1]} \qquad \text{Using (48) and (57)}
\end{aligned} \tag{58}$$

Taking $\epsilon$ to 0 in (58) leads to the desired lower bound (47). □

## B  Exponential-like Arms

In this section, we study extreme bandit problem with exponential-like arms, assuming that the reward distribution of each arm exhibits exponential decay near its maximum $b$:

$$P(X > x; a, b) \sim e^{-\frac{ab}{b-x}} \quad \text{with} \quad a > 0 \text{ and } b \leq 1. \tag{59}$$

Specifically, we consider modified exponential distribution, We have the following preliminary

**Theorem 7.** *Assume the reward of the arm follow the distribution:*

$$P(x; a, b) = 1 - e^{-\frac{ax}{b-x}} \quad \text{with} \quad a > 0 \text{ and } b \leq 1. \tag{60}$$

*For random variables $X_t \sim P$, we have*

$$b - \mathbb{E}[\max_{t=1}^{T} X_t] \geq \frac{ab/e}{\ln(T+1)} \tag{61}$$

*Proof.* For the distribution $P(x; a, b)$, the probability density function (PDF) is given by:

$$\rho(x; a, b) = \frac{ab}{(b-x)^2}e^{-\frac{ax}{b-x}} \qquad (0 \leq x \leq b).$$

The PDF of its maximum over $n$ samples is

$$\rho_n(x; a, b) = n\rho(x; a, b)P(x; a, b)^{n-1} \qquad (0 \leq x \leq b)$$

with expectation

$$\begin{aligned}
\mathbb{E}_{\rho_n}[x] &= n\int_0^b x\rho(x; a, b)P(x; a, b)^{n-1}dx \\
&= nb\int_0^1 \Big(1 - \frac{a}{a - \ln(1-y)}\Big)y^{n-1}dy \qquad (y = P(x; a, b)) \\
&= b - nb\int_0^1 \frac{a}{a - \ln(1-y)}y^{n-1}dy
\end{aligned} \tag{62}$$

By using the following inequality

$$n \int_0^1 \frac{a}{a - \ln(1-y)} y^{n-1} dy \geq n \int_0^{1-\frac{1}{n+1}} \frac{a}{a - \ln(1-y)} y^{n-1} dy$$

$$\geq n \int_0^{1-\frac{1}{n+1}} \frac{a}{a - \ln(\frac{1}{n+1})} y^{n-1} dy \qquad \text{replacing } y \text{ by } 1 - \frac{1}{n+1}$$

$$= \frac{a}{a - \ln(\frac{1}{n+1})} \left(1 - \frac{1}{n+1}\right)^n$$

$$\geq \frac{a/e}{\ln(n+1)} \quad \text{using } \left(1 - \frac{1}{n+1}\right)^n \geq \frac{1}{e}$$

$$\tag{63}$$

We have the following upper bound about the expectation

$$b - nb \int_0^1 \frac{a}{a - \ln(1-y)} y^{n-1} dy \leq b\left(1 - \frac{a/e}{\ln(n+1)}\right) \tag{64}$$

$$\square$$

By using Theorem 7, we have the performance gap satisfies

$$G(T) \geq b_1 - \max_{k=1}^K \mathbb{E}[\max_{t \leq T} X_{k,t}]$$

$$\geq \min_{k=1}^K \left\{ \Delta_k + \frac{a_k b_k}{e \ln(T+1)} \right\} \tag{65}$$

$$\geq \min\left\{ \frac{a_1 b_1}{e \ln(T+1)}, \min_{k \geq 2} \Delta_k \right\}.$$

## C  Numerical Experiments with Beta Distributions

In this section, we construct a simple simulated numerical environment to compare the performance of the UCB-extreme policy proposed in this paper (see (7)) with the classic $\epsilon$-greedy and UCB1 policies. Concurrently, we experimentally validate the theoretical upper bounds for the performance gap $G(T)$ and regret $R(T)$ derived in Theorem 1 (specifically, (9) and (12)).

We consider a multi-armed bandits problem with a total of $K = 4$ arms. The reward for each arm $k$ follows a Beta distribution with parameters $(a_k, b_k)$ (defined in (8)). The specific parameter settings are as follows:

- Arm 1: $a_1 = 3, b_1 = 1$
- Arm 2: $a_2 = 4, b_2 = 0.9$
- Arm 3: $a_3 = 2, b_3 = 0.85$
- Arm 4: $a_4 = 1, b_4 = 0.9$

We employ the three aforementioned policies to conduct a total of $T = 50,000$ independent sampling rounds on these four arms. The entire experiment is repeated $400$ times to obtain statistical averages, thereby mitigating the effects of randomness. It should be noted that since the constant $C$ obtained from (11) is relatively large, the theorem imposes a very high requirement on the number of steps in the numerical experiments. Here, we fix $C$ at 10.

The parameter configurations for the policies are as follows:

- For the $\epsilon$-greedy policy, we set the exploration rate $\epsilon = 0.25$.
- For the UCB-extreme policy, in accordance with the conditions in (10), we set the parameters $2c = \left(\frac{7}{3}\right)^{\frac{1}{3}}$ and $\gamma = \frac{1}{3}$.
- To ensure a fair comparison, the exploration parameters for the UCB1 policy are set to the same values as those for UCB-extreme.

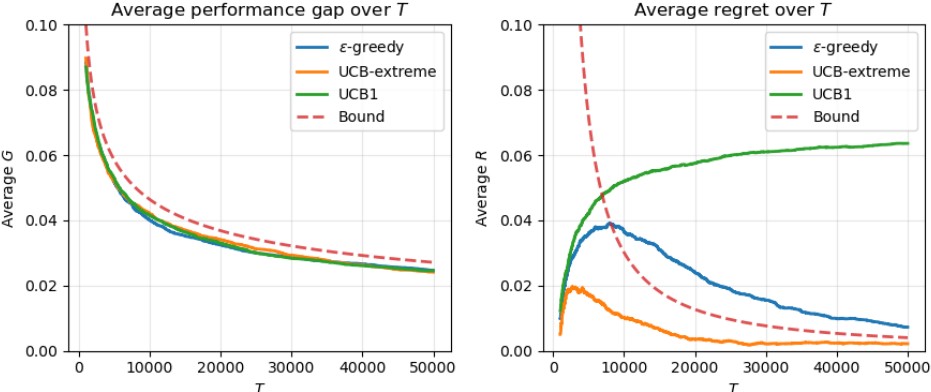

Figure 4: Numerical experimental results of three different strategies for $G(T)$ and $R(T)$, along with their corresponding upper bounds, are shown in the figures. The regret results on the right demonstrate that our UCB-extreme performs the best.

The experimental results are presented in Appendix C. It is clearly observable from the figure that, in this specific environment, only the $R(T)$ generated by our proposed UCB-extreme policy satisfies the upper bound (12). In contrast, the $R(T)$ for both the $\epsilon$-greedy and UCB1 policies exceeds this bound. Notably, the traditional UCB1 policy, which uses the sample mean as its expectation estimate, performs the worst, remaining far from the upper bound. Meanwhile, the $\epsilon$-greedy policy gradually approaches this bound in the later stages of the algorithm's execution.

On the other hand, regarding the performance gap $G(T)$, its definition is inherently independent of the specific policy employed. Consequently, the experimental results also confirm that $G(T)$ under all three policies adheres to the theoretical upper bound given by (9).

## D   Modeling Symbolic Regression as a Markov Decision Process

In this section we describe in detail how to cast the symbolic regression problem as a Markov Decision Process (MDP). First, an expression can be represented by a binary expression tree, and any such tree can be mapped to a symbol sequence via a traversal order. Here we use preorder traversal, as illustrated in Figure 1. Conversely, a symbol sequence that obeys the preorder rules can be reconstructed into an expression tree as follows:

1. Locate the deepest non-full operator node in the current partial tree.

2. Expand that node by adding a child symbol, first to the left and then to the right.

3. Repeat until there are no non-full operator nodes remaining.

This procedure yields a complete binary tree, which corresponds uniquely to an expression. In code, this can be implemented very efficiently using a stack. Note that, because we impose a depth limit on expressions, in some cases only terminal symbols (variables or constants) may be added.

So in practice, we can formulate the symbolic regression task as an MDP defined by the tuple

$$\mathcal{M} = \big(\mathcal{S},\ \mathcal{A},\ P,\ R,\ \beta\big),$$

where:

- $\mathcal{S}$ is the state space, consisting of all partial symbol sequences (or incomplete binary expression trees) whose depth does not exceed a maximum $H$. Formally,

$$\mathcal{S} = \big\{\, s = (a_1, \ldots, a_k) \ \big| \ \mathrm{depth}(E(s)) \leq H,\ s \text{ obeys valid preorder structure}\big\},$$

where $E(s)$ reconstructs the expression tree from $s$ and $\mathrm{depth}(\cdot)$ measures tree depth.

- $\mathcal{A}$ is the (state-dependent) action space. Let $\mathcal{O}$ be the set of operators, $\mathcal{X}$ the set of variables, and $\mathcal{C}$ the set of constants. At state $s_t$:

$$\mathcal{A}(s_t) = \begin{cases} \mathcal{O} \cup \mathcal{X} \cup \mathcal{C}, & \mathrm{depth}(E(s_t)) < H, \\ \mathcal{X} \cup \mathcal{C}, & \mathrm{depth}(E(s_t)) = H, \end{cases}$$

  i.e. when the partial tree is at maximum depth, actions are restricted to terminal symbols (variables or constants) to ensure the depth constraint.

- The transition function $P : \mathcal{S} \times \mathcal{A}(s) \to \mathcal{S}$ is deterministic: given $s_t = (a_1, \ldots, a_t)$ and action $a_{t+1} \in \mathcal{A}(s_t)$,

$$s_{t+1} = \mathrm{Expand}(s_t, a_{t+1}),$$

  where 'Expand' locates the deepest non-full operator node in the tree corresponding to $s_t$ and attaches $a_{t+1}$ as its next child (left first, then right).

- The reward function $R : \mathcal{S} \times \mathcal{A}(s) \to \mathbb{R}$ is zero at all nonterminal steps and gives a terminal payoff upon completion (no non-full operators remain):

$$R(s_t, a_{t+1}) = \begin{cases} 0, & s_{t+1} \text{ is non-terminal}, \\ \dfrac{1}{1 + \mathrm{NRMSE}\big(E(s_{t+1})\big)}, & s_{t+1} \text{ is terminal}, \end{cases}$$

  where $E(s)$ denotes the expression tree reconstructed from $s$ and NRMSE its normalized error on the training data.

- The discount factor $\beta \in [0, 1]$ is set to 1, since all reward is issued at termination, yielding return

$$G = \sum_{t=0}^{T-1} \beta^t R(s_t, a_{t+1}) = R(s_{T-1}, a_T).$$

An optimal policy $\pi^*$ maximizes the expected return

$$\pi^* = \arg\max_\pi \mathbb{E}[G \mid \pi],$$

which corresponds to finding the expression with minimal NRMSE. In practice, we explore this MDP via MCTS rollouts and apply genetic operators on the queue of promising sequences, as detailed in Algorithm 1.

## E Ablation Study

To isolate the contributions of our two key modifications to MCTS—the extreme-bandit node selection strategy and the evolution-inspired state-jumping actions—we perform an ablation study on the Nguyen benchmark. We evaluate these models:

1. **Model A**: Replaces the UCB-extreme selection rule with uniform random selection (i.e., set $\epsilon = 1$ in Table 4).

2. **Model B**: Model B: Omits the evolution-inspired state-jumping actions and only depends on the standard MCTS rollout to generate expressions(i.e., set $g_s = 0$ in Table 4).

3. **Model C**: MCTS using the standard UCB1 formula, not employing any methods proposed in this paper.

The experimental results in Table 2 show that both Model A and Model B perform worse than the complete algorithm, with Model B suffering a particularly large drop. This clearly highlights that the evolution-inspired state-jumping actions are the primary driver of our performance gains. At the same time, the UCB-extreme strategy continues to contribute positively, as its absence in Model A also leads to a noticeable degradation. In line with our previous analysis, the state-jumping moves dramatically reshape the reward landscape during the search process, allowing the algorithm to explore more promising regions of the expression space and to generate more complex yet high-quality expressions.

Simultaneously, we can also observe that Model C performs very poorly and shows a significant gap compared to Model B. This further demonstrates the effectiveness of UCB-extreme for symbolic regression problems.

Table 2: Recovery rate comparison (%) for 3 ablations on Nguyen benchmark.

| | Ours | Model A | Model B | Model C |
|---|---|---|---|---|
| Nguyen-1 | **100** | **100** | **100** | 4 |
| Nguyen-2 | **100** | **100** | 38 | 0 |
| Nguyen-3 | **100** | 98 | 7 | 0 |
| Nguyen-4 | **97** | 90 | 0 | 0 |
| Nguyen-5 | **100** | 89 | 41 | 0 |
| Nguyen-6 | **100** | **100** | **100** | 3 |
| Nguyen-7 | **100** | 99 | 53 | 0 |
| Nguyen-8 | **100** | **100** | 98 | 47 |
| Nguyen-9 | **100** | **100** | **100** | 1 |
| Nguyen-10 | **100** | **100** | **100** | 0 |
| Nguyen-11 | **100** | **100** | **100** | 71 |
| Nguyen-12* | **22** | 14 | 0 | 0 |
| Nguyen average | **93.25** | 90.83 | 53.08 | 10.50 |

Finally, we also conducted further hyperparameter tuning experiments based on Model B. This serves both to validate the theoretical insights behind our design and to provide a preliminary analysis of the characteristics of different symbolic regression problems. The details can be found in Appendix F.

## F   Reward Tail Decay Rate Analysis on Nguyen benchmark

We performed an empirical investigation of reward distributions across the Nguyen benchmark suite (see Figure 5). The majority of expressions demonstrate tail decay rates ranging approximately between 4 and 6, with notable exceptions observed in expressions 5 and 12. Special attention is required regarding our use of Nguyen-12 rather than Nguyen-12*(the latter was specifically employed for recovery-rate testing). Furthermore, the integration of state-jumping operations, drawing inspiration from genetic programming methodologies, yields significant enhancements to the global reward profile: expressions 5, 8, 11, and 12 display tail decay rates approximately within $[2, 4]$, whereas the remaining expressions predominantly exhibit values below 2.

Based on these observations, we carried out recovery-rate experiments under six distinct UCB-extreme parameter settings. All evaluations were conducted using the MCTS algorithm that adopts only the UCB-extreme strategy without the evolution-inspired state-jumping actions (i.e., with $p_s = 0$ and $\epsilon = 0$, as defined in Algorithm 1). This was done to better understand the difficulty of symbolic regression tasks and to validate our theoretical analysis. All experiments followed the evaluation protocol outlined in Appendix H.

Within our framework, decreasing the discount factor $\gamma$ or increasing the exploration constant $c$ both promote exploration. The six tested configurations, denoted as Models A–F, are defined as follows:

1. **Model A**: $\gamma = \frac{1}{2}, \quad 2c = \sqrt{2}$.
   Standard UCB1 parameters, used as a baseline.

2. **Model B**: $\gamma = \frac{1}{2}, \quad 2c = \sqrt{\frac{5}{2}}$.
   Same discount factor as Model A, with an increased $c$ corresponding to $a_1 = 2$.

3. **Model C**: $\gamma = \frac{1}{4}, \quad 2c = \left(\frac{9}{4}\right)^{1/4}$.
   Parameters calibrated to $a_1 = 4$, increasing exploration via reduced $\gamma$.

4. **Model D**: $\gamma = \frac{1}{6}, \quad 2c = \left(\frac{13}{6}\right)^{1/6}$.
   Configured for $a_1 = 6$, with further reduction in $\gamma$ and an adjusted $c$.

5. **Model E**: $\gamma = \frac{1}{8}, \quad 2c = \left(\frac{17}{8}\right)^{1/8}$.
   The most exploratory setting among Models A–E, corresponding to $a_1 = 8$.

Table 3: Recovery rate comparison (%) on the Nguyen benchmark of MCTS using only the UCB-extreme strategy under six different UCB-extreme parameter configurations.

|  | Model A | Model B | Model C | Model D | Model E | Model F |
|---|---|---|---|---|---|---|
| Nguyen-1 | 90 | 93 | **100** | **100** | **100** | **100** |
| Nguyen-2 | 29 | 37 | 77 | 96 | **100** | 92 |
| Nguyen-3 | 10 | 13 | 17 | 25 | **52** | 25 |
| Nguyen-4 | 0 | 2 | 0 | 1 | **7** | 0 |
| Nguyen-5 | 1 | 0 | 22 | 18 | 8 | **41** |
| Nguyen-6 | 49 | 73 | **100** | **100** | 99 | **100** |
| Nguyen-7 | 27 | 30 | 89 | **98** | 72 | **98** |
| Nguyen-8 | 16 | 12 | 69 | 96 | 95 | **100** |
| Nguyen-9 | 56 | 66 | **100** | **100** | **100** | **100** |
| Nguyen-10 | 35 | 39 | **100** | **100** | **100** | **100** |
| Nguyen-11 | 57 | 59 | 99 | **100** | **100** | **100** |
| Nguyen-12* | 0 | 0 | 0 | 0 | 0 | **1** |
| Nguyen average | 30.83 | 35.33 | 64.42 | 69.50 | 69.42 | **71.42** |

6. **Model F**: $\gamma = \frac{1}{8}, \quad 2c = \left(\frac{26}{3}\right)^{1/8}$.

Constructed by fixing $a_1 = 6$ and $\gamma = \frac{1}{8}$, while enforcing equality only in the second inequality of (10).

Models B–E are constructed by fixing $a_1 = 2, 4, 6, 8$, respectively, and then selecting $\gamma$ and $c$ such that both inequalities in (10) are satisfied with equality. This calibration strategy ensures that each configuration corresponds to a specific theoretical tail decay rate. In contrast, Model F enforces equality only in the second inequality of (10), thereby relaxing the first constraint.

From a theoretical standpoint, any reward distribution conforming to the polynomial decay form discussed in Appendix A with $6 \leq a_1 \leq 8$ should satisfy (10) under the parameterization of Model F. Empirically, Model F yields the highest recovery rates, with detailed results reported in Table 3. These findings suggest improved performance when $a_1 \geq 6$, as confirmed by the quantitative metrics in the referenced table.

Moreover, Table 3 indicates that different equations benefit from different parameter configurations for optimal performance. We also explored alternative settings where $\gamma$ was held fixed while $c$ was incrementally increased. However, these variants did not yield statistically significant improvements in performance. On the contrary, excessively large values of $c$ were found to degrade solution quality, underscoring the importance of balanced parameter calibration.

## G  Algorithm Details

This section provides a detailed account of the core algorithmic components. As previously discussed, each node in the Monte Carlo Tree Search (MCTS) framework maintains an independent trajectory queue. This functionality is supported by two complementary mechanisms: *Backward Propagation* (Algorithm 2), which propagates complete trajectories upward through ancestor nodes, and *Forward Propagation* (Algorithm 3), which disseminates partial trajectories downward along promising branches.

It is important to highlight that the selection strategy in Algorithm 1 is parametrically flexible—it can be configured to reduce to either an $\epsilon$-greedy policy or a pure UCB-extreme strategy (7), depending on the parameter settings. Our hybrid approach demonstrates superior performance in symbolic regression tasks, as evidenced by the results in Section 4.

In contrast to conventional genetic programming, which relies on dedicated selection mechanisms (e.g., tournament selection or roulette-wheel selection) to maintain population diversity, our frame-

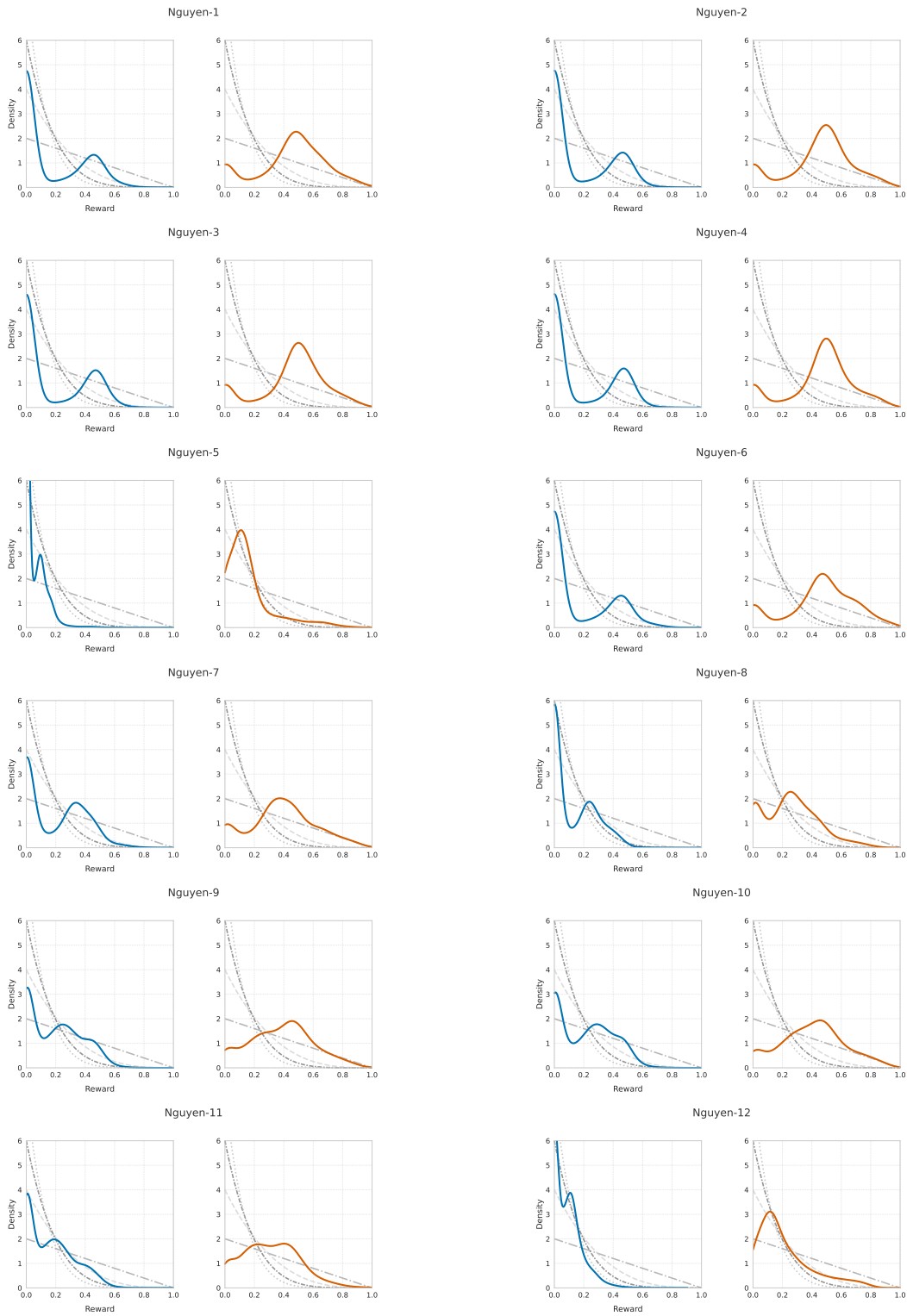

Figure 5: Empirical reward distributions on the Nguyen benchmarks with $\epsilon = 1$ in Algorithm 1 (fully random node selection) and a budget of 200,000 expression evaluations. Other settings follow Table 4. Results from 100 runs are shown using Gaussian KDE (bandwidth 0.25). The blue and orange curves represent standard MCTS simulation and the state-jumping actions, respectively. Overlaid gray curves depict beta distributions defined on the interval [0,1], parameterized by tail decay rates with $a = 2, 4, 6, 8$.

**Algorithm 2:** Backward Propagation

**Input:** Current node $v_0$, trajectory $\tau = (a_1, \ldots, a_m)$
**Output:** Updated trajectory queues $\{\mathcal{Q}_v\}$

1 $r \leftarrow \text{COMPUTEREWARD}(\tau)$
2 $v \leftarrow v_0$          ▷ Start from current node
3 **while** $v \neq \text{null}$ **do**          ▷ Traverse ancestors
4     **if** $\text{ENQUEUE}(\mathcal{Q}_v,\ \tau,\ r)$ **= False**: **break**          ▷ Abort if enqueue fails
5     $\tau \leftarrow [a_v] \,\|\, \tau$          ▷ Prepend action to trajectory
6     $v \leftarrow v.\text{parent}$          ▷ Move to parent node
7 **end while**

---

**Algorithm 3:** Forward Propagation

**Input:** Current node $v_0$, trajectory $\tau = (a_1, \ldots, a_m)$
**Output:** Updated trajectory queues $\{\mathcal{Q}_v\}$

1 $r \leftarrow \text{COMPUTEREWARD}(\tau)$
2 $v \leftarrow v_0$          ▷ Start from current node
3 **for** $i \leftarrow 1$ **to** $m$ **do**          ▷ Follow trajectory
4     $v_{\text{next}} \leftarrow \{u \in \mathcal{C}(v) \mid a_u = a_i\}$          ▷ Find child with matching action
5     **if** $v_{\text{next}} = \emptyset$: **break**          ▷ Stop if path deviates
6     $v \leftarrow v_{\text{next}}$          ▷ Move to next node
7     $\text{ENQUEUE}(\mathcal{Q}_v,\ \tau_{i+1:m},\ r)$          ▷ Store remaining suffix
8 **end for**

---

work adopts a streamlined Top-$N$ selection operator to fulfill a similar role with competitive effectiveness. Moreover, the integration of MCTS with bidirectional propagation enables the systematic retention of high-quality expressions across different exploration paths. This design not only preserves high-reward solutions but also enhances the ability to escape local optima through informed traversal of the state space.

## H  Experimental Details

All experiments were conducted on machines delivering 10.6 TFLOPS of FP32 compute performance and 256GB RAM.

### H.1  Algorithm Parameters

The hyperparameter configurations used in the comparative study are summarized in Table 4. Note that while the values of $p_s$, $\epsilon$, and the maximum expression evaluation budget vary in Appendix F, all other settings and experimental conditions remain consistent.

Crossover is implemented using single-point crossover, where two expression trees exchange subtrees at randomly selected nodes, with each node chosen with equal probability. Mutation operations include uniform mutation, node replacement, subtree insertion, and subtree shrinkage, each applied with equal probability.

### H.2  Metrics and Procedures

**Basic Benchmarks.** For the ground-truth benchmarks, we adopt the recovery rate metric as defined in [16]. The recovery rate is the proportion of trials in which the original expression is successfully rediscovered, evaluated over 100 trials with fixed random seeds. Each trial is subject to a maximum of 2 million expression evaluations.

To ensure fair comparison, we adopt the same action space constraints as in [16], [23], and [27]:

- A trigonometric function may not have another trigonometric function as a descendant.
- An `exp` node must not be followed by a `log` node, and vice versa.

Table 4: Hyperparameters of Algorithm 1 used in the comparative study. Unless otherwise specified, all experimental settings follow the configuration listed in this table.

| Hyperparameter Name | Symbol | Value | Comment |
|---|---|---|---|
| **Expression Parameters** | | | |
| Maximum expression depth | - | 6 | - |
| Maximum expression constants | - | 10 | No limit for SRBench |
| Available Symbol Set | - | $\{+, -, *, /, \sin, \cos,$ $\exp, \log, \text{constant}, \text{variable}\}$ | Omit constant for Nguyen & Livermore |
| **Algorithm Parameters** | | | |
| UCB-extreme parameter 1 | $c$ | 1 | - |
| UCB-extreme parameter 2 | $\gamma$ | 0.5 | - |
| Size of queue | $N$ | 500 | - |
| State-jumping rate | $g_s$ | 0.2 | - |
| Mutation rate | $g_m$ | 0.1 | - |
| Random explore rate | $\epsilon$ | 0.2 | - |

Additional structural constraints are listed in Table 4. Any expression generated through crossover or mutation that violates these constraints is discarded and does not contribute to the evaluation budget.

**SRBench Black-box Benchmarks.** For black-box benchmarks, we follow the evaluation protocol of [10]. Each dataset is split into training and testing subsets (75%/25%) using a fixed random seed. We perform 10 independent runs per dataset, each constrained to a maximum of 500,000 evaluations or 48 hours, whichever occurs first. Performance is reported as the median test $R^2$ score across the 10 runs:

$$R^2 = 1 - \frac{\sum_{i=1}^{n}(y_i - \hat{y}_i)^2}{\sum_{i=1}^{n}(y_i - \bar{y})^2}, \quad \text{Complexity} = \big|\mathcal{T}(f(\cdot))\big|,$$

where $\bar{y}$ denotes the mean of the true output values. The `Complexity` term represents the number of nodes in the simplified expression tree, as computed using `Sympy` [56].

### H.3 Short Descriptions of Baselines for Basic Benchmarks

Below is a concise overview of the baseline algorithms used in our Basic Benchmarks:

- **DSR** [16]: Deep Symbolic Regression (DSR) employs a recurrent neural network (RNN) to sample candidate expressions and a risk-seeking policy gradient algorithm to iteratively update network parameters, thereby steering the sampling distribution toward high-reward symbolic forms.

- **GEGL** [53]: Genetic Expert-Guided Learning (GEGL) generates $M$ candidate expressions via RNN, ranks them by reward, and selects the Top-$N$ candidates. Genetic programming operators are applied to these elites to produce offspring, which are subsequently re-ranked to retain the Top-$N$ evolved variants. The RNN is then updated through imitation learning on the combined set of original and evolved Top-$N$ expressions.

- **NGGP** [23]: Neural-Guided Genetic Programming (NGGP) enhances DSR by seeding the genetic programming population with RNN-sampled expressions. It subsequently fine-tunes the RNN via risk-seeking policy gradient updates based on the best solutions discovered during the GP run, establishing a bidirectional feedback loop between neural sampling and evolutionary optimization.

- **PySR** [15]: PySR is an open-source, high-performance genetic programming library featuring multi-population evolutionary strategies. Implemented in Python and Julia, it efficiently discovers compact symbolic models through parallelized evolutionary search.

In our experimental configuration, PySR's core parameters were set as follows: `maxdepth` $= 6$, `populations` $= 20$, `niterations` $= 100$, and `population_size` $= 1000$. Notably, we deliber-

ately abstained from constraining PySR's action space, as [16] substantiates that such limitations adversely affect genetic programming efficacy. All other baseline algorithms strictly adhered to the parameterizations detailed in [23]. The symbol set configuration matches the specifications documented in Table 4.

## H.4  Detailed Experimental Results

Table 5 reports the recovery rates obtained on each equation of the Nguyen and Livermore benchmarks. Table 6 presents a comparison of single-core runtimes on the Nguyen benchmark between our algorithm and NGGP. The reported runtimes represent the average over 100 independent runs for each equation, ensuring statistical reliability. These results indicate that our algorithm achieves recovery rates comparable to those of NGGP while demonstrating substantially faster runtimes and requiring fewer equation evaluations to recover each target expression.

## H.5  Specifications of Basic Benchmarks

All ground-truth benchmark configurations are listed in Table 7. The notation $U(a, b, c)$ denotes drawing $c$ independent uniform samples from the interval $[a, b]$ for each input variable.

Table 5: Recovery rate comparison (%) across Nguyen and Livermore benchmarks.

(a) Nguyen benchmark

|  | Ours | DSR | GEGL | NGGP | PySR |
|---|---|---|---|---|---|
| Nguyen-1 | **100** | **100** | **100** | **100** | **100** |
| Nguyen-2 | **100** | **100** | **100** | **100** | 98 |
| Nguyen-3 | **100** | **100** | **100** | **100** | 70 |
| Nguyen-4 | 97 | **100** | **100** | **100** | 2 |
| Nguyen-5 | **100** | 72 | 92 | **100** | 89 |
| Nguyen-6 | **100** | **100** | **100** | **100** | **100** |
| Nguyen-7 | **100** | 35 | 48 | 96 | 35 |
| Nguyen-8 | **100** | 96 | **100** | **100** | 99 |
| Nguyen-9 | **100** | **100** | **100** | **100** | **100** |
| Nguyen-10 | **100** | **100** | 92 | **100** | **100** |
| Nguyen-11 | **100** | **100** | **100** | **100** | **100** |
| Nguyen-12* | **22** | 0 | 0 | 12 | 0 |
| Average | **93.25** | 83.58 | 86.00 | 92.33 | 74.41 |

(b) Livermore benchmark

|  | Ours | DSR | GEGL | NGGP | PySR |
|---|---|---|---|---|---|
| Livermore-1 | **100** | 3 | **100** | **100** | **100** |
| Livermore-2 | 95 | 87 | 44 | **100** | 32 |
| Livermore-3 | **100** | 66 | **100** | **100** | 97 |
| Livermore-4 | **100** | 76 | **100** | **100** | 90 |
| Livermore-5 | **78** | 0 | 0 | 4 | 30 |
| Livermore-6 | 19 | 97 | 64 | **88** | 0 |
| Livermore-7 | **15** | 0 | 0 | 0 | 0 |
| Livermore-8 | **22** | 0 | 0 | 0 | 0 |
| Livermore-9 | 2 | 0 | 12 | **24** | 0 |
| Livermore-10 | **35** | 0 | 0 | 24 | 26 |
| Livermore-11 | **100** | 17 | 92 | **100** | 92 |
| Livermore-12 | 91 | 61 | **100** | **100** | 18 |
| Livermore-13 | **100** | 55 | 84 | **100** | **100** |
| Livermore-14 | **100** | 0 | **100** | **100** | 15 |
| Livermore-15 | **100** | 0 | 96 | **100** | 74 |
| Livermore-16 | 72 | 4 | 12 | 92 | **93** |
| Livermore-17 | 69 | 0 | 4 | 68 | **72** |
| Livermore-18 | 29 | 0 | 0 | **56** | 3 |
| Livermore-19 | **100** | **100** | **100** | **100** | 88 |
| Livermore-20 | **100** | 98 | **100** | **100** | 76 |
| Livermore-21 | 41 | 2 | **64** | 24 | 0 |
| Livermore-22 | **100** | 3 | 68 | 84 | 9 |
| Average | **71.41** | 30.41 | 56.36 | 71.09 | 46.14 |

Table 6: Comparison of average single-core runtimes (in seconds) over 100 independent runs between our algorithm and NGGP on the Nguyen benchmark.

|  | Ours | NGGP |
|---|---|---|
| Nguyen-1 | **6.63** | 27.05 |
| Nguyen-2 | **29.10** | 59.79 |
| Nguyen-3 | **105.84** | 151.06 |
| Nguyen-4 | **254.03** | 268.88 |
| Nguyen-5 | **93.00** | 501.65 |
| Nguyen-6 | **16.46** | 43.96 |
| Nguyen-7 | **49.68** | 752.32 |
| Nguyen-8 | **99.41** | 123.21 |
| Nguyen-9 | **9.82** | 31.17 |
| Nguyen-10 | **71.06** | 103.72 |
| Nguyen-11 | **2.99** | 66.50 |
| Nguyen-12* | 1156.70 | **1057.11** |
| Average | **157.90** | 265.54 |

Table 7: Specifications of Basic Benchmarks. It should be noted that configurations marked with an asterisk (*) indicate the use of the input domain sampled from the interval [0, 10], consistent with the experimental framework in [23].

| Name | Expression | Dataset |
|------|-----------|---------|
| Nguyen-1 | $x^3 + x^2 + x$ | $U(-1, 1, 20)$ |
| Nguyen-2 | $x^4 + x^3 + x^2 + x$ | $U(-1, 1, 20)$ |
| Nguyen-3 | $x^5 + x^4 + x^3 + x^2 + x$ | $U(-1, 1, 20)$ |
| Nguyen-4 | $x^6 + x^5 + x^4 + x^3 + x^2 + x$ | $U(-1, 1, 20)$ |
| Nguyen-5 | $\sin(x^2)\cos(x) - 1$ | $U(-1, 1, 20)$ |
| Nguyen-6 | $\sin(x) + \sin(x + x^2)$ | $U(-1, 1, 20)$ |
| Nguyen-7 | $\log(x + 1) + \log(x^2 + 1)$ | $U(0, 2, 20)$ |
| Nguyen-8 | $\sqrt{x}$ | $U(0, 4, 20)$ |
| Nguyen-9 | $\sin(x) + \sin(y^2)$ | $U(0, 1, 20)$ |
| Nguyen-10 | $2\sin(x)\cos(y)$ | $U(0, 1, 20)$ |
| Nguyen-11 | $x^y$ | $U(0, 1, 20)$ |
| Nguyen-12 | $x^4 - x^3 + \frac{1}{2}y^2 - y$ | $U(0, 1, 20)$ |
| Nguyen-12* | $x^4 - x^3 + \frac{1}{2}y^2 - y$ | $U(0, 10, 20)$ |
| Livermore-1 | $\frac{1}{3} + x + \sin(x^2)$ | $U(-10, 10, 1000)$ |
| Livermore-2 | $\sin(x^2)\cos(x) - 2$ | $U(-1, 1, 20)$ |
| Livermore-3 | $\sin(x^3)\cos(x^2) - 1$ | $U(-1, 1, 20)$ |
| Livermore-4 | $\log(x + 1) + \log(x^2 + 1) + \log(x)$ | $U(0, 2, 20)$ |
| Livermore-5 | $x^4 - x^3 + x^2 - y$ | $U(0, 1, 20)$ |
| Livermore-6 | $4x^4 + 3x^3 + 2x^2 + x$ | $U(-1, 1, 20)$ |
| Livermore-7 | $\sinh(x)$ | $U(-1, 1, 20)$ |
| Livermore-8 | $\cosh(x)$ | $U(-1, 1, 20)$ |
| Livermore-9 | $x^9 + x^8 + x^7 + x^6 + x^5 + x^4 + x^3 + x^2 + x$ | $U(-1, 1, 20)$ |
| Livermore-10 | $6\sin(x)\cos(y)$ | $U(0, 1, 20)$ |
| Livermore-11 | $\frac{x^2 y^2}{x + y}$ | $U(-1, 1, 20)$ |
| Livermore-12 | $\frac{x^5}{y^3}$ | $U(-1, 1, 20)$ |
| Livermore-13 | $x^{\frac{1}{3}}$ | $U(0, 4, 20)$ |
| Livermore-14 | $x^3 + x^2 + x + \sin(x) + \sin(x^2)$ | $U(-1, 1, 20)$ |
| Livermore-15 | $x^{\frac{1}{5}}$ | $U(0, 4, 20)$ |
| Livermore-16 | $x^{\frac{2}{5}}$ | $U(0, 4, 20)$ |
| Livermore-17 | $4\sin(x)\cos(y)$ | $U(0, 1, 20)$ |
| Livermore-18 | $\sin(x^2)\cos(x) - 5$ | $U(-1, 1, 20)$ |
| Livermore-19 | $x^5 + x^4 + x^2 + x$ | $U(-1, 1, 20)$ |
| Livermore-20 | $\exp(-x^2)$ | $U(-1, 1, 20)$ |
| Livermore-21 | $x^8 + x^7 + x^6 + x^5 + x^4 + x^3 + x^2 + x$ | $U(-1, 1, 20)$ |
| Livermore-22 | $\exp(-0.5x^2)$ | $U(-1, 1, 20)$ |
| Nguyen-1[c] | $3.39x^3 + 2.12x^2 + 1.78x$ | $U(-1, 1, 20)$ |
| Nguyen-5[c] | $\sin(x^2)\cos(x) - 0.75$ | $U(-1, 1, 20)$ |
| Nguyen-7[c] | $\log(x + 1.4) + \log(x^2 + 1.3)$ | $U(0, 2, 20)$ |
| Nguyen-8[c] | $\sqrt{1.23x}$ | $U(0, 4, 20)$ |
| Nguyen-10[c] | $\sin(1.5x)\cos(0.5y)$ | $U(0, 1, 20)$ |
| Jin-1 | $2.5x^4 - 1.3x^3 + 0.5y^2 - 1.7y$ | $U(-3, 3, 100)$ |
| Jin-2 | $8.0x^2 + 8.0y^3 - 15.0$ | $U(-3, 3, 100)$ |
| Jin-3 | $0.2x^3 + 0.5y^3 - 1.2y - 0.5x$ | $U(-3, 3, 100)$ |
| Jin-4 | $1.5\exp(x) + 5.0\cos(y)$ | $U(-3, 3, 100)$ |
| Jin-5 | $6.0\sin(x)\cos(y)$ | $U(-3, 3, 100)$ |
| Jin-6 | $1.35xy + 5.5\sin((x - 1.0)(y - 1.0))$ | $U(-3, 3, 100)$ |

