# OpenReview forum: "Improving Monte Carlo Tree Search for Symbolic Regression"
_NeurIPS.cc/2025/Conference — NeurIPS 2025 poster_

### Official Review · Reviewer_SL53 · 2025-07-01

**Clarity:** 3
**Significance:** 2
**Originality:** 2
**Rating:** 4
**Confidence:** 4

**Summary:**

This paper proposes an improved Monte Carlo Tree Search (MCTS) framework for symbolic regression. To address the challenge of efficiently finding the global optimal symbolic expression, the framework introduces two key innovations: the first is the extreme bandit allocation strategy, which models the node selection process as a K-arm extreme bandit problem, uses a specific formula to balance exploration and exploitation, and theoretically guarantees convergence to the optimal solution in a finite time by analyzing the upper and lower bounds of the performance gap and regret value. The second is the state jump action inspired by evolution, which introduces genetic operations such as mutation and crossover, and combines bidirectional propagation to achieve non-local jumps in the search process, reshape the reward distribution, and improve the efficiency of exploring high-reward areas. The framework is based on Markov decision process (MDP) modeling and is verified by experiments on benchmark datasets such as Nguyen and Livermore.

**Questions:**

- Is the extreme bandit allocation strategy similar to the following paper? In this paper, the regret upper bounds under stationary distribution and non-stationary distribution are considered respectively. What is the difference between you two?

    Such as, [1] Hu Y Q, Liu X H, Li S Q, et al. Cascaded algorithm selection with extreme-region UCB bandit[J]. IEEE Transactions on Pattern Analysis and Machine Intelligence, 2021, 44(10): 6782-6794.

- The derivation premise of the article requires that the reward distribution has a known polynomial tail decay characteristic. This assumption is very strong. Will it lead to situations where it is not applicable in some scenarios?

- In Appendix Table 2, it can be seen that the experimental results are extremely sensitive to hyperparameters and have poor robustness. The performance is highly dependent on the choice of two parameters. The recovery rate under different combinations varies significantly, and some combinations are almost ineffective. This shows that vanilla MCTS lacks generalization ability. In practical applications, it is difficult to find a set of universal parameters that adapt to all tasks. So is the purpose of this experiment to find the best parameter experience? Or is there another purpose? Are the parameter settings in model F used in subsequent experiments?

- The state jumping mechanism introduced by this method significantly changes the tail shape of the reward distribution, thereby improving the search efficiency. However, the change of the reward landscape is static during the execution of the algorithm. Is it possible to design a dynamic and adaptive state jumping strategy that automatically adjusts the jump probability or operation type according to the current search stage, thereby further improving the convergence and generalization ability of the search?

**Ethical Concerns:**

["NO or VERY MINOR ethics concerns only"]

**Final Justification:**

N/A

**Limitations:**

- The extreme bandit strategy relies on empirical parameters. Is it possible to introduce adaptive tuning or reward-tail-aware automatic calibration mechanism to reduce parameter dependence?

- All performance bounds and regret derivations are based on the assumption that rewards follow a specific form of beta-like distribution with a polynomial decay in the tail. I would like to know if this makes sense and why.

**Quality:**

2

**Strengths And Weaknesses:**

**Strengths:**

- An extreme bandit allocation strategy suitable for extreme value optimization is proposed, and the state jump mechanism in the evolutionary algorithm is organically integrated into the MCTS framework, which significantly improves the search efficiency and the ability to jump out of the local optimum.

- The upper and lower bounds of the performance of the strategy under the polynomial tail distribution are systematically derived, providing solid theoretical support for the method and filling the theoretical gap of the existing MCTS in symbolic regression.

**Weaknesses:**

- See Questions and Limitations.

---

> ### Author Rebuttal · Authors · 2025-07-30
>
> Dear Reviewer,
>
> Thank you very much for your careful review of our paper and for your valuable comments. We have read all of your questions carefully and respond to each point below.
>
> ---
>
> **Question 1** Relation to Hu et al. [1] “Extreme‑Region UCB Bandits”
>
> **Response**: We appreciate you highlighting Hu et al. (IEEE TPAMI 2021) and have added this reference to Section 1.1. While both works use bandits to find the highest-reward configuration (differing from traditional cumulative-reward bandits), key distinctions exist:
>
> 1. **Theoretical Objective and Assumptions**
>
>    - **Objective**: The primary objective of Hu et al. is to maximize the extreme-region, with a regret based on the probability of that region. In contrast, our work seeks to maximize the maximum reward of the bandit directly, with regret defined by this maximum reward. We believe our objective is better suited for symbolic regression, as the goal of the task is to find the best possible expression (yielding the maximum reward).
>
>    - **Assumption**: Extreme multi-armed bandit systems often require strong assumptions about the reward distribution. For example, in a bandit with a small probability of a very large reward, the maximum reward can be difficult to find in a finite number of trials. Hu et al. assume the reward distribution satisfies a specific moment condition:
>
>      $$ \ln \mathbb{E}[e^{\lambda|X - EX|}] \geq \psi(\lambda) \textrm{,where } \psi \textrm{ is convex}$$
>
>      Our work, however, studies reward distributions with polynomial or exponential-like tails. We found that for polynomial-like tails, our designed allocation strategy is effective, achieving an optimal regret of $\mathcal{O}(\frac{\ln T}{T^{1+1/a_1}})$, where $a_1$ is the polynomial tail decay rate. Conversely, for exponential-like tails, the problem becomes unsolvable and requires an exponential number of trials. This is consistent with the fact that symbolic regression is an NP-hard problem.
>
> 2. **Practical Usage**
>
>    - Both Hu et al's extreme-bandit and our approach have hyperparameters. In our method, one only needs to approximate the polynomial-tail decay rate, $a_1$. The hyperparameters can then be selected based on our theoretical result in Equation (10):
>
>      $$ \frac{1}{\gamma} \geq a_1 \qquad \textrm{and} \qquad 2^{a_1}\cdot a_1^{\frac{1}{\gamma}} \geq 2 + \frac{1}{a_1}. $$
>
>    - A sensitivity study of these hyperparameters is presented in Appendix C Table 2, where symbolic regression problems with polynomial-tail reward distribution with $a_1 \in (4,6)$. Table 2 shows any set of hyperparameters satisfying Equation (10), specifically $\gamma \leq \frac{1}{4}$, yields similarly good results. However, as we mentioned, if the tail is not polynomial-like, the problem may not be solvable.
>
>    - We also conducted experiments comparing our approach with Hu et al's `ER-UCB-S` strategy for symbolic regression, with the results listed below:
>
>    **Recovery rate comparison (%) on Nguyen benchmark.**
>    | | **Ours** | **ER-UCB-S*** | **UCB-extreme*** |
>    | :--- | :---: | :---: | :---: |
>    | **Nguyen average** | **93.25** | 44.00 | 71.42 |
>
>    The other two methods in the table serve as baselines, generating expressions using only the MCTS rollout process without our evolutionary state-jumping actions. The `UCB-extreme` methods utilize the allocation strategy from our paper for direct comparison. The asterisk (*) indicates tuned parameters. Our method consistently performed better, likely due to the reward distributions' tails aligning with a polynomial-tail distribution ($a_1 = 2$), as shown in Figures 2 and 4. It is also possible that the parameters for `ER-UCB-S` were not optimally tuned. Symbolic regression algorithms typically use maximum reward in their allocation strategies [2].
>
> ---
>
> **Question 2** The derivation premise of the article requires that the reward distribution has a known polynomial tail decay characteristic. This assumption is very strong. Will it lead to situations where it is not applicable in some scenarios?
>
> **Response**: We understand your concern about the strong assumption of a known polynomial-tail decay. Our algorithm uses this as a heuristic, fitting the decay rate $a_1$ and applying improved MCTS. This assumption is reasonable because:
>
> - A larger $a_1$ reflects greater sparsity of high-reward expressions, matching practical observations and capturing the "rare optimal-solution" phenomenon in symbolic regression.
> - Empirical evidence (Figures 2 and 4) supports this pattern, showing that state-jump actions flatten the tail decay and improve benchmark performance. For the Nguyen dataset, the tail is well-approximated by a polynomial decay ($a_1 = 2$), enabling effective parameter settings ($\gamma = \frac{1}{2}$, $c = 1$).
>
> However, exceptions exist, such as Nguyen-12, where the reward distribution approximates an exponential decay tail, making the problem nearly unsolvable, as shown by the 0% recovery rate in Table 2. This result aligns with our theoretical findings that exponential decay tails pose significant challenges for symbolic regression.
>
> ---
>
> **Question 3** Appendix C Table 2, Sensitive to Hyperparameters and have poor robustness
>
> **Response**: Thank you for your careful review. We believe there may have been a misunderstanding regarding the parameter study in Appendix C, and we appreciate the opportunity to clarify our findings.
>
> Appendix C, Table 2, validates our theoretical results about the extreme-bandit allocation strategy rather than finding the "best" parameters. For optimal regret, hyperparameters must satisfy Equation (10), specifically $
> \gamma \leq \frac{1}{a_1}$. Tests on the Nguyen benchmark show that Models A and B with $
> \gamma = \frac{1}{2}$ violate Equation (10), leading to poor performance. Models C, D, E, and F approximately satisfy Equation (10), demonstrating robustness and acceptable results.
>
> It is important to note that Appendix C experiments do not incorporate state-jump actions, which significantly improve the reward distribution (shifting from blue to orange curves in Figure 4). This improvement typically brings the polynomial decay rate closer to $a_1 = 2$ for the Nguyen benchmark. Appendix C’s experiments were not meant to find the "best" parameters, and Model F’s parameters were **not** used for final comparisons. Instead, we used parameters corresponding to a reward distribution with $a_1 = 2$ for the main text comparisons.
>
> Finally, Appendix C, Table 2, uses MCTS only with our extreme-bandit allocation strategy. Our improved MCTS framework, described in the main paper, is significantly more robust as it combines this allocation strategy with state-jump actions and random exploration probability.
>
> ---
>
> **Question 4** Dynamic, Adaptive State-Jumping Strategy
>
> **Response**: Thank you for this insightful suggestion! Designing a dynamic exploring procedure for MCTS is a promising future direction. While we did not adopt this approach in our current work, we conducted preliminary tests on two strategies: `linear-decrease` and `linear-increase`. The `linear-decrease` strategy reduces state-jumping probability over time, whereas the `linear-increase` strategy does the opposite. The results are listed below:
>
>    **Recovery rate comparison of dynamic strategies (%) on the Nguyen benchmark.**
>    | | **Ours(Static)** | **Linear-decrease** | **Linear-increase** |
>    | :--- | :---: | :---: | :---: |
>    | **Nguyen average** | 93.25 | 92.58 | **93.50** |
>
>
> ---
>
> **Limitations 1** Adaptive tuning or reward-tail-aware automatic calibration mechanism
>
> **Response**: We thank the reviewer for the suggestion. Our extreme-bandit allocation strategy relies on a key parameter $a_1$, the polynomial-tail decay rate of the reward distribution. Then, we can compute the hyperparameters based on our theoretical results in Equation (10). We can set $\gamma = \frac{1}{a_1}$ and $c = \frac{a_1}{2}$, and this is also how we determine these hyperparameters used in the present work.
>
> We designed a simple adaptive strategy for the UCB-extreme parameters. During the search, we dynamically adjusted the estimation of $a_1$ based on the gap between the maximum sampled reward and the optimal reward: increasing it when the gap was large and decreasing it when the gap was small. The parameters $\gamma$ and $c$ were updated according to the changing estimate of $a_1$. The results are as follows:
>
>    **Recovery rate comparison of an adaptive parameter strategy (%) on the Nguyen benchmark.**
>
>    | | **Ours (Static)** | **Adaptive-UCB** |
>    | :--- | :---: | :---: |
>    | **Nguyen average** | **93.25** | 92.75 |
>
> The results indicate that this adaptive heuristic performs slightly worse than the static setting. While automatic calibration is promising, this implementation may be too simplistic for complex search dynamics. Future work could explore first estimating $a_1$ from the rewards and then automatically calculating the UCB-extreme parameters $\gamma$ and $c$.
>
> ---
>
> **Limitations 2** Meaning and Necessity of the Tail Assumption
>
> **Response**: This limitation is addressed in our response to Question 2.
>
> ---
>
> Once again, we appreciate your detailed feedback. We believe our paper offers significant theoretical and practical innovations by uniting extreme-bandit theory with evolutionary heuristics in MCTS for Symbolic Regression. We will revise the manuscript to incorporate your suggestions and sincerely hope you will reconsider your evaluation.
>
> Sincerely,
> The Authors
>
> ---
>
> **Reference**
> [1] Hu Y. Q., Liu X. H., Li S. Q., et al. "Cascaded Algorithm Selection with Extreme‑Region UCB Bandit" *IEEE Transactions on Pattern Analysis and Machine Intelligence*, 2021, 44(10): 6782–6794.
>
> [2] Shojaee P, Meidani K, Barati Farimani A, et al. "Transformer-based planning for symbolic regression"[J]. *Advances in Neural Information Processing Systems*, 2023, 36: 45907-45919.

---

> > ### Comment · Reviewer_SL53 · 2025-08-05
> > **Response to [Rebuttal by Authors]**
> >
> > I have carefully read your response to the reviewers' comments and thank you for your thoughtful response and detailed explanation of the relevant issues. Your response clearly demonstrates your grasp of research details and your attention to the reviewers' comments. The relevant explanations and supplementary information have fully addressed the questions raised previously.

---

> > > ### Author Response · Authors · 2025-08-05
> > >
> > > Dear Reviewer,
> > >
> > > Thank you very much for your encouraging feedback and for taking the time to carefully review our response.
> > >
> > > ---
> > >
> > > We are pleased to know that the explanations and supplementary information have addressed the questions raised previously. If there are any additional points or suggestions, we would be more than happy to address them.
> > >
> > > ---
> > >
> > > Thank you once again for your valuable time and thoughtful suggestions.
> > >
> > > Sincerely,
> > > The Authors

---

### Official Review · Reviewer_3PJv · 2025-07-02

**Clarity:** 3
**Significance:** 3
**Originality:** 3
**Rating:** 5
**Confidence:** 2

**Summary:**

This study proposes a new Monte Carlo Tree Search (MCTS)-based method for symbolic regression.
Particularly, two technical contributions are made: i) an extreme bandit allocation strategy for identifying globally optimal expressions with a theoretical guarantee, and ii) state-jumping actions embedded in MCTS. While standard MCTS pursues maximizing expected cumulative rewards, this work adapts MCTS to the case of highest reward maximization. Numerical experiments support the validity of the proposed approach and also demonstrate that the proposed method achieves competitive scores compared with state-of-the-art methods.

**Questions:**

The questions come from the weaknesses except for the last one.
1. Is the polynomial-tail assumption intuitively reasonable or just made artificially?
2. Is it possible to create an artificial dataset or task for which we have better control over reward distribution to further analyze the derived bounds?
3. Does each of the new MCTS and mutation contribute sufficiently to the symbolic regression performance?
4. In Theorem 1, it is assumed that the first arm is optimal. Can this assumption be made without loss of generality? If not, how does this assumption become reasonable?

**Ethical Concerns:**

["NO or VERY MINOR ethics concerns only"]

**Final Justification:**

A new MCTS method that adapts to the extreme bandit case is proposed. The theoretical analysis provides an upper bound on the regret with an assumption on reward density.

Through the rebuttal and discussion, the polynomial tail assumption is more justified by the pointers to and clarification of experiments (both in the main text and the appendix). The ablation also demonstrates that each of the two proposed techniques makes a sufficient contribution.

The paper has solid theoretical analysis, which is also supported by experiments. Thus, I recommend this work for acceptance.

**Limitations:**

Yes.

**Paper Formatting Concerns:**

No.

**Quality:**

3

**Strengths And Weaknesses:**

**Strengths**
- A new MCTS method that adapts to the extreme bandit case is proposed. The theoretical analysis provides an upper bound on the regret with an assumption on reward density.
- Inspired by genetic programming, state-jumping actions are introduced to the MCTS framework. Unlike prior methods that, for example, alternate between genetic programming and reinforcement learning, the proposed method rigorously embeds mutations into MCTS.
- The experiments analyze the validity of the assumption on reward density. The impact of hyperparameters is also extensively examined. Finally, in symbolic regression performance, the proposed method outperforms all baselines on Basic Benchmarks, while it also performs competitively with state-of-the-art methods on the SRBench benchmark.

---

**Weaknesses**

I feel that this work makes solid contributions from both theoretical and practical points of view, although I'm not an expert in this field. I suggest the following points to improve this work further.

**1. Discussion on the assumption.**
Theorems 1/3 present upper/lower bounds on the regret assuming the distribution of rewards has polynomial/exponential tail. While I appreciate these positive and negative claims, it would be better if the authors could elaborate more on these assumptions. Is the polynomial-tail assumption intuitively reasonable or just made artificially? Figure 2 provides empirical justification, but further analysis may give insights. Is it possible to create an artificial dataset or task for which we have better control over reward distribution to further analyze the derived bounds?

**2. Ablation.**
There are two proposed components, one about MCTS and another about mutations. However, it is unclear from the experiments how much performance gain was obtained from each in the final performance.

**3. Minor comments**
- [l.255] It was initially unclear what range [4, 6] refers to. It would be better to clarify that this is the range of the first argument of the Beta distribution.

---

> ### Author Rebuttal · Authors · 2025-07-30
>
> Dear Reviewer,
>
> Thank you for your thorough review and valuable comments. We have provided a point-by-point response below, which we hope will address all of your concerns. The manuscript has been revised accordingly.
>
> ---
>
>
> **Question 1** Is the polynomial-tail assumption intuitively reasonable or just made artificially?
>
> **Response**: The polynomial-tail assumption is introduced to provide a theoretical foundation for our guarantees. We think this assumption is reasonable for the following reasons:
>
> - The tail parameter $a_1$ controls how rapidly the reward distribution decays near its maximum. A larger $a_1$ reflects greater sparsity of high-reward expressions, matching practical observations. This perfectly captures the "rare optimal-solution" phenomenon inherent in the symbolic regression problem.
>
> - Empirically, Figures 2 and 4 (blue curve on the right) provide strong support for this reward-distribution pattern. Furthermore, our state-jump actions (Figures 2 and 4) flatten the tail decay and significantly improve benchmark performance (orange curve on the right). Moreover, our assumptions are empirically verifiable. For the Nguyen dataset, the tail (with state-jump actions) is well-approximated by a polynomial decay with $a_1 = 2$. We use this empirical estimation to set $\gamma = \frac{1}{2}$ and $c = 1$, based on Equation (10) in our theory. This single set of parameters was used for all the over 100 tests in this work and yielded excellent results.
>
> We also acknowledge that for certain cases, such as the Nguyen-12 expression $x^4 - x^3 + \frac{1}{2}y^2 - y$, the reward distribution does not follow a polynomial decay tail but instead approximates an exponential decay tail. As a result, nearly all of our attempts for this case, as reported in Table 2, achieved a 0% recovery rate. This exception supports our theoretical finding that an exponential decay tail makes the problem nearly unsolvable with our current approach.
>
> While we believe this assumption is robust for many practical scenarios, we agree that future work should explore relaxing it to accommodate a broader class of distributions, and we have noted this in our manuscript.
>
> ---
>
> **Question 2** Is it possible to create an artificial dataset or task for which we have better control over reward distribution to further analyze the derived bounds?
>
> **Response**: Thank you for this excellent and insightful suggestion. To further validate our theoretical bounds, we have conducted a series of experiments using multiple Beta distributions to simulate controlled reward environments. In these tests, we compared the performance of our UCB-extreme strategy against standard baselines, including UCB1 and epsilon-greedy, by analyzing their performance gap $G(T)$ and regret $R(T)$.
>
> The performance gap, $G(T)$, is by definition independent of the allocation strategy. As expected, our experiments confirmed that $G(T)$ satisfies the bound in Equation (9) of Theorem 1. However, the regret, $R(T)$, is policy-dependent and serves as the key differentiator. The empirical results demonstrate that only the regret of UCB-extreme consistently satisfies the theoretical upper bound in Equation (12) of Theorem 1. In contrast, strategies like UCB1 and epsilon-greedy, which are not designed for extreme value optimization, failed to meet these guarantees. This outcome highlights the superiority of our allocation strategy in efficiently identifying the maximal reward.
>
> As we cannot include images in this response, we will add the detailed results and visualizations for these experiments to the appendix.
>
> We also agree that creating an artificial dataset with controllable reward distributions would be a significant contribution to the field of symbolic regression. Such a dataset would provide a controlled environment to systematically test theoretical bounds and analyze algorithm behavior under diverse conditions. It could also serve as a foundation for developing standardized metrics to quantify task difficulty—a much-needed tool for the community. We believe this is a promising direction for future research and will add this point to the future work section of our manuscript.
>
> ---
>
> **Question 3** Does each of the new MCTS and mutation contribute sufficiently to the symbolic regression performance?
> **Response**: We agree that a formal ablation study is essential. While our initial submission demonstrated strong performance, it did not formally quantify each component’s independent effect. To address this, we have now conducted an ablation study with three distinct models:
>
> - Model A: Replaces the UCB-extreme selection rule with uniform random selection.
> - Model B: Omits the evolution-inspired state-jumping actions and only relies on the standard MCTS rollout to generate expressions.
> - Model C: MCTS using the standard UCB1 formula, without any of the innovations proposed in this paper.
>
> The results for these models are summarized below:
>
> **Recovery rate comparison (%) for 3 ablations on Nguyen benchmark.**
> | | **Ours** | **Model A** | **Model B** | **Model C** |
> | :--- | :---: | :---: | :---: | :---: |
> | Nguyen-1 | **100** | **100** | **100** | 4|
> | Nguyen-2 | **100** | **100** | 38 | 0|
> | Nguyen-3 | **100** | 98 | 7 | 0|
> | Nguyen-4 | **97** | 90 | 0 | 0|
> | Nguyen-5 | **100** | 89 | 41 | 0|
> | Nguyen-6 | **100** | **100** | **100** | 3|
> | Nguyen-7 | **100** | 99 | 53 | 0|
> | Nguyen-8 | **100** | **100** | 98 | 47|
> | Nguyen-9 | **100** | **100** | **100** | 1|
> | Nguyen-10 | **100** | **100** | **100** |0 |
> | Nguyen-11 | **100** | **100** | **100** | 71|
> | Nguyen-12* | **22** | 14 | 0 | 0|
> | **Nguyen average** | **93.25** | 90.83 | 53.08 | 10.50|
>
>
> The results clearly demonstrate the contribution of each component.
> - **Model C**, which uses the standard UCB1, performs poorly, confirming that vanilla MCTS is insufficient for this complex search problem.
> - **Model B** shows a dramatic performance drop compared to our full model, highlighting that the **evolution-inspired state-jumping actions are the most critical innovation**. These actions enable the algorithm to effectively explore and escape local optima, and we attribute their effectiveness to a tree-like multi-queue design that provides strong selection pressure while preserving a large pool of high-quality expressions.
> - **Model A** performs better than Model B but is still outperformed by our full model, which indicates that the **UCB-extreme selection strategy provides a significant and complementary benefit** by focusing the search on high-reward regions.
>
> In our revision, we will add the detailed results and experimental setup for this study to the appendix.
>
> ---
>
>
> **Question 4** In Theorem 1, it is assumed that the first arm is optimal. Can this assumption be made without loss of generality? If not, how does this assumption become reasonable?
> **Response**: In Theorem 1, we assume that a particular arm (e.g., the first arm) is optimal for notational simplicity; this does not restrict the generality of the result.  If multiple arms are optimal, or if the optimal arm is not indexed first, one can simply relabel or reindex the arms and apply the same analysis. Thus, the assumption is benign and does not limit the theorem’s applicability.
>
> ---
>
>
> **Clarification of “Range [4, 6]”**
>
> **Response**: Thank you for pointing this out. We will clarify the meaning of “range [4, 6]” in the revised manuscript and provide a more detailed explanation in the main text.
>
> ---
>
> Once again, thank you for your insightful feedback, which has not only helped us improve the manuscript but also inspired several promising directions for our future research. We believe our revisions have addressed your concerns and look forward to your evaluation.
>
> Sincerely,
> The Authors

---

> > ### Comment · Reviewer_3PJv · 2025-08-01
> > **Response from Reviewer**
> >
> > Thank you for your clarification and elaboration. Overall, the rebuttal addresses my concerns. While I'll keep my eyes on the interactions between the authors and reviewers (particularly, Reviewer SL53), I'd keep my original score unless any critical weaknesses are revealed/unsolved.
> >
> > - [R1]. I see that the polynomial-tail assumption is essential in this work. The proposed framework requires a set of hyperparameters to model this assumption. The state-jumping actions indeed flatten the decay (the authors refer to the blue curve tail following a=6 and the orange one following \beta=2 in Fig. 2, right? - at first glance, the tails were not literally *flattened*, which made me a bit confused though). The failure in Nguyen-12 matches the statement on exponential decay. While there are several practical concerns regarding the choice of these hyperparameters, the authors provide theoretical bounds to determine them practically, and the experiments validate their utility.
> > - [R2]. Thank you for conducting a controlled experiment. The empirical superiority over the baseline seems to support the theoretical claims.
> > - [R3]. Thank you for adding the ablation. The contribution from the proposed components is now clearer.

---

> > > ### Author Response · Authors · 2025-08-03
> > >
> > > Dear Reviewer,
> > >
> > > Thank you very much for your positive and constructive feedback. We are delighted to hear that our rebuttal has addressed your main concerns.
> > >
> > > ---
> > >
> > > We also appreciate your observation regarding our use of the term "flattened" for the tails in Figure 2, and we apologize for any confusion our imprecise wording may have caused. You are absolutely right that the tails are not literally horizontal. A more precise description, would be that the state-jumping actions cause the tail's decay rate to become significantly slower (i.e., a polynomial decay).
> > >
> > > We are glad that the new controlled experiment and ablation study have helped clarify our contributions and their empirical support. We will do our best to constructively address all points raised by the other reviewers.
> > >
> > > ---
> > >
> > > We are sincerely grateful for your valuable time and insightful feedback, which have been instrumental in improving our manuscript.
> > >
> > > Sincerely,
> > > The Authors

---

### Official Review · Reviewer_hpnf · 2025-07-04

**Clarity:** 2
**Significance:** 3
**Originality:** 3
**Rating:** 4
**Confidence:** 4

**Summary:**

This paper introduces new variants of Monte Carlo Tree Search (MCTS) for the symbolic regression (SR) problem. Compared to the standard approach, the authors propose (1) an extreme-bandit allocation strategy and provide theoretical optimality guarantees under certain assumptions, and (2) evolution-inspired state-jumping actions. The proposed methods are evaluated against state-of-the-art baselines on common benchmarks.

**Questions:**

Please see the weaknesses section above.

**Ethical Concerns:**

["NO or VERY MINOR ethics concerns only"]

**Limitations:**

Some core components of the proposed SR methods are not clearly explained. The paper is good, but the issues raised above need to be addressed for the paper to be acceptable.

**Paper Formatting Concerns:**

None.

**Quality:**

3

**Strengths And Weaknesses:**

Strengths
-----------
* The proposed approaches are well-motivated and address relevant challenges in SR.
* The experimental results are strong and demonstrate the effectiveness of the methods.
* The heuristic designs for mutation and crossover operations appear reasonable and practically sound.

Weaknesses
----------------
Some core components of the proposed SR methods are not clearly explained. In more detail,

* For the extreme-bandit strategy, the authors state that the objective is to maximize J(T), as the gap between the maximum possible reward and the expected highest reward, and hence the regret. It is unclear why the goal is to maximize rather than minimize J(T), and the paper does not provide sufficient justification for this design choice.

* There is little information about the exact actions (i.e., which symbols) used when modeling the MDP. How are numerical expressions in SR generated? Are they also produced via MCTS? The paper provides limited details about these aspects of the SR formulation, even in the appendix.

---

> ### Author Rebuttal · Authors · 2025-07-30
>
> Dear Reviewer,
>
> Thank you very much for your careful review of our paper and your valuable feedback. We have read all of your comments carefully and respond to your concerns point by point below.
>
> ---
>
> **Question 1** Clarification of the Optimization Objective (Minimizing $J(T)$)
>
> **Response**: Thank you for pointing this out. We apologize for the confusion caused by a typographical error in the manuscript. The objective should indeed be to **minimize** $J(T)$, which measures the regret as the gap between the maximum possible reward and the expected highest reward obtained. As described in the paper, $J(T)$ can be decomposed into two parts: $G(T)$ and $R(T)$, and since $G(T)$ is independent of the policy, the optimization reduces to minimizing $R(T)$. Specifically, the word **"maximizes"** in line 159 should be corrected to **"minimizes,"** and **"maximizing"** in line 165 should be corrected to **"minimizing."** We appreciate your careful attention to this detail.
>
> ---
>
> **Question 2** Clarification of MDP Modeling and Action Space in Symbolic Regression
>
> **Response**: We acknowledge that the original version of the paper did not explain the MDP modeling details clearly enough. Here is a more comprehensive explanation, which we plan to add to the Appendix in the revised version.
>
> An expression can be naturally represented as a binary tree (see Figure 1 in the paper). By conducting a preorder traversal of the binary tree, we obtain a symbol sequence. Conversely, a valid symbol sequence can be converted into a binary expression tree using the following rule:
>    1. Locate the deepest operator node that does not yet have the required number of children.
>    2. Expand this node by appending children in a left-then-right order.
>    3. Continue until all nodes are complete and the full tree is constructed.
>
> This procedure can be efficiently implemented using a stack.
>
> Accordingly, the symbolic regression task can be formulated as an MDP defined by the tuple $\mathcal{M} = (\mathcal{S}, \mathcal{A}, P, R, \beta)$, where:
>
> - **State** $\mathcal{S}$: The state space consists of all partial symbol sequences that represent incomplete binary expression trees, constrained by a maximum depth $H$. Formally, a state is $s = (a_1, \dots, a_k)$ such that the reconstructed tree $E(s)$ satisfies $\mathrm{depth}(E(s)) \le H$ and the sequence maintains a valid preorder structure.
>
> - **Action** $\mathcal{A}$: The action space is state-dependent. Let $\mathcal{O}$ be the set of operators, $\mathcal{X}$ the set of variables, and $\mathcal{C}$ the set of constants. At a given state $s_t$, the available actions are:
>   $$
>   \mathcal{A}(s_t) = \mathcal{O} \cup \mathcal{X} \cup \mathcal{C}, \quad \text{if } \mathrm{depth}(E(s_t)) < H
>   $$
>   $$
>   \mathcal{A}(s_t) = \mathcal{X} \cup \mathcal{C}, \quad \text{if } \mathrm{depth}(E(s_t)) = H
>   $$
>   This means that when the tree reaches its maximum depth, subsequent actions are restricted to terminal symbols (variables or constants) to ensure the depth constraint is not violated.
>
> - **Transition** $P$: The transition function $P(s_{t+1}|s_t, a_{t+1})$ is deterministic. Given a state $s_t$ and an action $a_{t+1}$, the next state is generated by appending the new symbol: $s_{t+1} = (s_t, a_{t+1})$. This corresponds to expanding the deepest, not-yet-full operator node in the expression tree $E(s_t)$ by adding $a_{t+1}$ as its next child (left-to-right), as mentioned above.
>
> - **Reward** $R$: The reward function provides a signal only upon the completion of a valid expression. For all intermediate steps, the reward is 0. A terminal reward is given when the sequence $s_{t+1}$ corresponds to a complete expression tree:
>   $$
>   R(s_t, a_{t+1}) = 0, \quad \text{if } s_{t+1} \text{ is non-terminal}
>   $$
>   $$
>   R(s_t, a_{t+1}) = \frac{1}{1 + \mathrm{NRMSE}(E(s_{t+1}))}, \quad \text{if } s_{t+1} \text{ is terminal}
>   $$
>   Here, NRMSE denotes the Normalized Root Mean Square Error of the final expression on the training data.
>
> - **Discount Factor** $\beta$: We set $\beta=1$ because rewards are only dispensed at the final step. The total return $G$ for an episode is simply the terminal reward.
>
> The goal is to find an optimal policy $\pi^* = \arg\max_{\pi} \mathbb{E}[G \mid \pi]$, which corresponds to discovering an expression with the minimum NRMSE.
>
> In our algorithm, expressions can be generated via random rollouts in MCTS or through mutation and crossover operations applied to sequences stored in the population queue, as described in Algorithm 1.
>
> ---
>
> Once again, we sincerely appreciate your thoughtful and constructive comments. They have helped us significantly improve the clarity of our paper. We will revise the manuscript accordingly and respectfully ask for your continued consideration of our work.
>
> Sincerely,
> The Authors

---

### Note · Authors · 2025-08-12

Dear Reviewers and Area Chairs,

First and foremost, we would like to express our sincere gratitude for your careful review and thoughtful feedback on our paper. We have endeavored to make a meaningful contribution to the field of symbolic regression and hope you find value in our efforts.

- Our work is the first to introduce extreme bandit theory into symbolic regression, for which we design an effective and concise allocation strategy, and provide theoretical analysis that opens up a new perspective for addressing the exploration-exploitation dilemma in this field.
- We propose an evolution‑inspired state‑jumping heuristic that integrates genetic algorithms with MCTS, offering a paradigm distinct from prior hybrid approaches.
- We demonstrate the effectiveness and strong performance of our method across multiple symbolic regression benchmarks.

During the rebuttal phase, we provided detailed clarifications and supplementary materials for all the points raised by the reviewers, which we were pleased to see were met with positive feedback.

Following the rebuttal, we conducted additional analyses and will incorporate the following revisions into the manuscript:
- Revised the text to improve clarity and readability and to reduce potential ambiguity.
- Added to the Appendix a detailed description of how symbolic regression can be formulated as a reinforcement learning problem.
- Validated our bounds with controlled Beta‑distributed rewards: G(T) matches predictions, and only UCB‑extreme regret consistently satisfies the theoretical upper bound.
- Added a focused ablation: state‑jumping is the most critical component; UCB‑extreme provides complementary gains.

Looking ahead, we plan to:
- Further validate the effectiveness of our UCB‑extreme allocation strategy—building on the preliminary comparisons in the rebuttal (e.g., against  UCB1 and Hu et al.’s ER‑UCB‑S)—and explore deeper refinements to improve its sampling efficiency.
- Further investigate dynamic state‑jump schedules and automatic parameter calibration; during the rebuttal, we explored a few simple preliminary experiments that suggested some promise.

Taken together, these updates strengthen the clarity, theoretical grounding, and empirical evidence of our approach. We would be grateful for your continued consideration and we also welcome any further suggestions and remain committed to improving the work.

Thank you once again for your valuable time and professional review.

Sincerely,
The Authors

---

### Decision · Program_Chairs · 2025-09-17

**Decision:**

Accept (poster)

**Comment:**

This paper introduces new variants of Monte Carlo Tree Search (MCTS) for the symbolic regression (SR) problem. Compared to the standard approach, the authors propose (1) an extreme-bandit allocation strategy and provide theoretical optimality guarantees under certain assumptions, and (2) evolution-inspired state-jumping actions. The proposed methods are evaluated against state-of-the-art baselines on common benchmarks.

The paper has made significant contributions in (1) The proposed approaches are well-motivated and address relevant challenges in SR. In particular, an extreme bandit allocation strategy suitable for extreme value optimization is integrated into SR. (2) The experimental results are strong and demonstrate the effectiveness of the methods. (3) The heuristic designs for mutation and crossover operations appear reasonable and practically sound.

At the same time, the reviewers raise questions regarding the experimental comparisons and the ablation study. The most important aspect is whether the conditions used in theoretical analysis are widely applicable in a real-world context. The authors are encouraged to take into account the reviewers' feedback.